**METHOD**

# BindVAE: Dirichlet variational autoencoders for de novo motif discovery from accessible chromatin

Meghana Kshirsagar[1*] , Han Yuan[2] , Juan Lavista Ferres[1] and Christina Leslie[3*]

*Correspondence:
meghana.kshirsagar@microsoft.com;
cleslie@cbio.mskcc.org
[1] Microsoft, AI for Good Research
Lab, Redmond, WA, USA
[2] Calico Life Sciences, South San
Francisco, CA, USA
[3] Memorial Sloan Kettering Cancer
Center, New York, NY, USA

**Abstract**

We present a novel unsupervised deep learning approach called BindVAE, based on Dirichlet variational autoencoders, for jointly decoding multiple TF binding signals from open chromatin regions. BindVAE can disentangle an input DNA sequence into distinct latent factors that encode cell-type specific in vivo binding signals for individual TFs, composite patterns for TFs involved in cooperative binding, and genomic context surrounding the binding sites. On the task of retrieving the motifs of expressed TFs in a given cell type, BindVAE is competitive with existing motif discovery approaches.

**Keywords:** ATAC-seq, Transcription factor, Dirichlet, Variational autoencoders, VAE, k-mers, Motifs

## Background

The advent of the assay for transposase-accessible chromatin using sequencing (ATAC-seq) [1] and, more recently, its single-cell counterpart, scATAC-seq [2], have brought about the current ubiquity of chromatin accessibility data across numerous cell types from diverse organisms, tissue samples, and disease states. Chromatin accessibility maps various kinds of genomic elements, including regulatory elements such as gene promoters and intronic and intergenic enhancers that are occupied by transcription factors (TFs) as well as structural elements such as CTCF and cohesin binding sites that may anchor 3D chromatin loops. The DNA sequence signals underlying regions of open chromatin are therefore complex: while a single assay allows us to create an atlas of tens of thousands of accessible "peaks" in a given cell type, we expect that dozens of TFs occupy overlapping subsets of these peaks due to the presence of their cognate binding sites or those of cofactors. A key problem in regulatory genomics is interpreting the regulatory information encoded in all chromatin accessible peaks, namely TF binding sites and the TF-specific "regulatory grammars" that allow TFs to bind at different locations within the same peak.

Traditional methods for identifying TF binding sites in chromatin accessible regions involve performing searches and enrichment analyses with a library of known TF motifs, each encoded as a position-specific weight matrix (PWMs). These standard approaches are useful in finding strong signals but are confounded by the problem of redundant or missing motifs, the near-identity of motifs for closely related factors, and the inherent limitation of using weight matrices to define binding sites when more subtle binding sequence signals may be present. De novo motif discovery can be underpowered when the sequence signal is complex; for example, if an important TF binds a small fraction of accessible sites, enrichment-based motif discovery may fail to identify the corresponding binding motif.

To address the limitations of PWMs, a range of supervised machine learning methods using k-mer representations have been used to train sequence models to predict or decipher chromatin accessibility. The first such methods were k-mer based SVMs [3, 4], which accurately discriminate between accessible sites and negative (flanking or random genomic) sequences but are more difficult to interpret in terms of constituent TF signals; feature attribution methods have recently been introduced to extract explanatory sequence patterns from gapped k-mer SVM models [5]. Other approaches include SeqGL [6], which trains a group lasso logistic regression model on ATAC-seq data, where k-mer groups correspond to TF binding patterns; BindSpace [7], a latent semantic embedding method for TF SELEX-seq data that enables multi-class identification of the TF signals in genomic sequences; and a topic model approach based on discovering combinatorial binding of TFs from ChIP-seq data [8].

In parallel work, a range of deep learning models have been applied to chromatin accessibility and other epigenomic data sets. Popular methods use a one-hot encoding of DNA sequence and train convolutional neural networks (CNNs) [9–12] to predict epigenomic signals. While these methods have made impressive strides, there is still an interpretability issue, especially for chromatin accessibility data, which contains numerous binding patterns for distinct motifs, as opposed to TF binding data (e.g., ChIP-seq, ChIP-nexus, CUT&RUN), where one might hope to identify a smaller number of binding patterns for the targeted TF as well as its cofactors. Even in this latter setting, a complex process of extracting sequence patterns through feature attribution and aggregating them into motifs may be required for interpretation [13].

Broadly speaking, the regulatory genomics field uses chromatin accessibility as measured by ATAC-seq to map candidate gene regulatory elements, including enhancer elements for genes. We note that not all accessible elements are enhancers, and not all elements involved in gene regulation are accessible (e.g., repressive elements may not be associated with open chromatin). Acknowledging these caveats, the next steps in decoding gene regulatory programs are to (i) associate ATAC-seq peaks with target genes and (ii) identify the TFs that might be binding peak regions and therefore regulating target genes. Problem (i) is not addressed here but is the subject of a wide range of current research activities, including the use of chromosome conformation capture assays to map 3D promoter-enhancer interactions and of single-cell multiomic data to enable correlation of peak accessibility and gene expression across individual cells. Problem (ii) is what we address here, by decoding the TF binding signals in specific candidate regulatory elements (peaks). Solving both problems will lead to mechanistic insight into the TF networks that regulate individual genes and gene expression programs.

In this work, we develop a deep learning approach based on Dirichlet variational autoencoders (VAE) for modeling chromatin accessibility data, using a k-mer representation of genomic sequences as input (Fig. 1a). VAEs are a family of machine learning models that learn probability distributions with latent variables. Similar to autoencoders, they learn representations of the input data by compressing the input via a 'bottleneck' layer in the neural network. A VAE achieves this compression in a *probabilistic* manner, whereby the encoder transforms the input *x* into *parameters* describing a probability distribution, which it samples from to get the latent representation *z*. The decoder then reconstructs the input from the latent representation *z*, with the goal of making the output *x′* as close as possible to the input *x* by minimizing the reconstruction error. In the Dirichlet VAE or 'topic model' setting, we assume that the input bag of k-mers from a peak is generated by multiple 'topics' (we refer to individual components of the latent space as

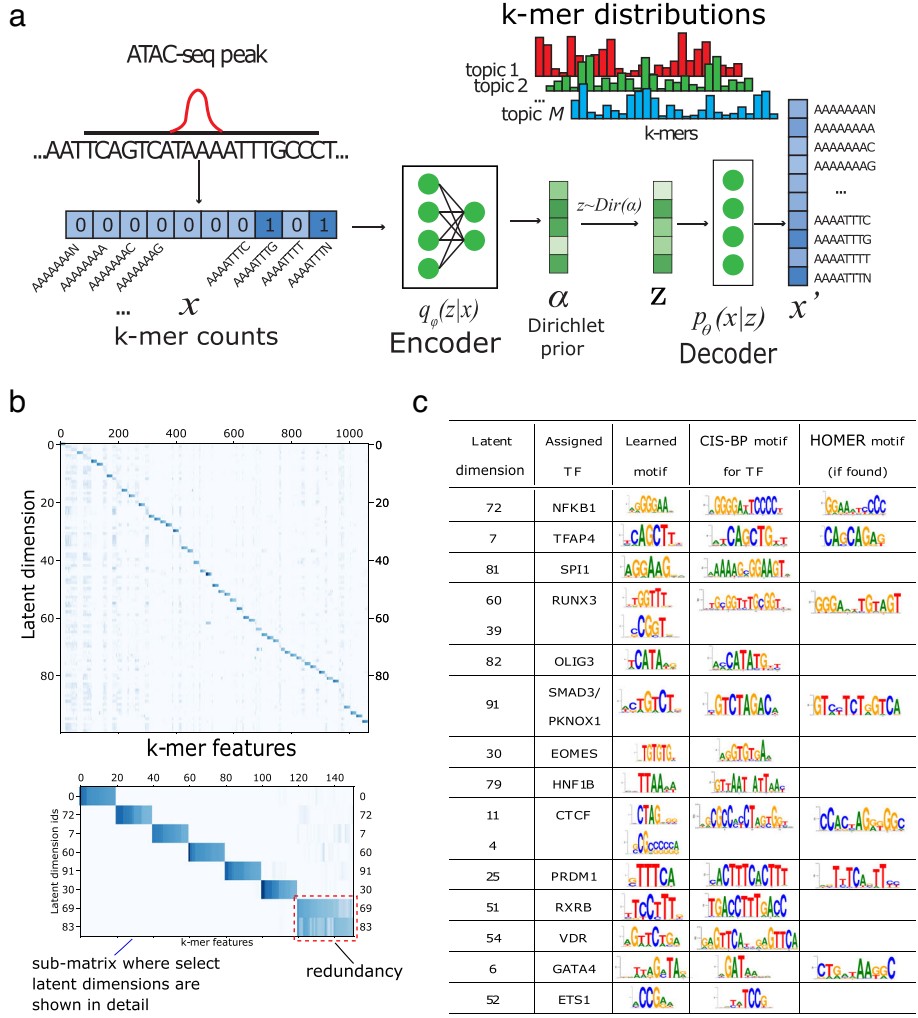

**Fig. 1** **a** Generative model captured by BindVAE. **b** The decoder weights $\theta \in \mathbb{R}^{100 \times 112800}$ from the GM12878 model, where each row *i* contains the 8-mer distribution learned for the latent dimension *i*. Since visualizing the weights of all 112800 8-mers is difficult, the weights for a subset of the 8-mers are shown. This subset of 8-mers was picked to contain the top 20 8-mers of each latent dimension. See the text for details. **c** Motif analysis of each latent dimension shows that different binding specificities are captured by each latent factor in GM12878. Column 2 shows the name of the TF assigned by Algorithm 1, column 3 shows the CIS-BP motif corresponding to the TF, and the last column shows the motif that was found by HOMER using de novo motif discovery for the TF

'latent dimension', 'topic' or 'latent factor'), which can correspond to binding signatures of TFs or other sequence signals in the data. The VAE formulation of latent variable models uses advances in neural network learning and enables efficient training on large data sets using backpropagation of gradients.

Here we show that the Dirichlet VAE model captures a useful representation of chromatin accessible elements, where the k-mer distributions encoded in the latent space can often be interpreted as binding patterns for TFs. We further present an algorithm to interpret the latent space that uses HT-SELEX probes. We show that our model learns cooperative binding signals for pairs of TFs, and we find that our model learns different TFs for distinct cell types, consistent with whether a TF is expressed or not.

## Results

### BindVAE: a Dirichlet variational autoencoder to deconvolve sequence signals

Each input example to BindVAE is the bag of DNA k-mers in one chromatin accessible region as shown in Fig. 1a. We describe our k-mer representation in detail in the "Methods" section. The generative model underlying the VAE is based on the observation that each peak is a combination of DNA sequence patterns from the following categories: (a) binding sites for one or more TFs; (b) low complexity regions; (c) genomic background; and (d) cleavage bias from the enzyme used in DNA fragmentation (or tagmentation). We thus surmise that the representation $z$, learned for each peak, should have latent dimensions (topics) that correspond to these categories, and in particular we are interested in topics that correspond to the binding signals of individual TFs. We assume that the membership of the peak in these categories follows a Dirichlet distribution. That is, $z_i \sim Dirichlet(\alpha)$. Each category (topic) in turn is represented as a multinomial distribution over k-mers. Therefore, we can think of a TF-related topic as representing a more general model of the TF's binding preferences, with the highest probability k-mers in the multinomial corresponding to the preferred binding signals. In Fig. 1a, the latent dimension topic 1 (red color), parameterized by $\theta_1$, contributes k-mers that capture the binding preferences of $TF_1$, while the blue-colored topic 2 might contribute k-mers representative of the genomic background. Note that the k-mer occurrences in a given DNA sequence are not independent, due to the overlap in the k-mers at successive locations of the input DNA sequence – the true generative process is thus not exactly multinomial. However, multinomial distributions have been successfully used in language models too, where the words that appear in a sentence are usually not independent of each other.

In general we find that the topics learned are not distributed mixtures of different signals: rather each latent component tends to capture a coherent pattern of k-mers for a single TF. This is a key advantage since, given an input DNA sequence, our model can *disentangle* it into dimensions containing TF-specific binding patterns and dimensions containing other types of content surrounding the binding sites.

We use k-mers with wildcards in our representation, meaning that we extend the alphabet of nucleotides with a wildcard character that matches all bases. This modeling choice is motivated by our previous k-mer based machine learning works including BindSpace [7], SeqGL [6], and our early string kernel work for epigenomic data [3] that anticipated the widely-used gkm-SVM method [14]. All these approaches use short k-mers (e.g. k=8) with some kind of inexact matching, such as matching to wildcards.

Because TF binding signals are degenerate, the use of wildcards that capture a larger number of binding instances has consistently proven useful in machine learning models of regulatory sequences.

The dimension of our latent space or the width of the bottleneck layer, which we call '$M$', is 100. In this way, the model encodes 100 topics, each corresponding to the binding signal of a TF or another enriched sequence pattern. For downstream quantitative evaluation and analysis, we mainly consider the 100-dimensional latent representations of inputs and the learned decoder parameters $\theta$, which guide the reconstruction of the input from the latent vector $z$. Details on VAEs, Dirichlet VAEs, the training approach we use, and hyper-parameter tuning are provided in the "Methods" section.

In the following sections, we present qualitative and quantitative analysis of the VAE models learned on ATAC-seq peaks from: GM12878, a human B lymphoblastoid cell line; A549, a lung epithelial cell line; and three other cell types discussed in Additional file 1: Table S1. We show the sequence motifs learned for TFs by summarizing the various dimensions of the latent space, project DNA sequences from other assays into the latent space for interpretation, locate cooperative binding signals, and correlate learned TFs to expression data.

### BindVAE learns diverse k-mer binding patterns

To summarize the DNA sequence patterns captured by BindVAE, we visualize the learned weights of the top 20 k-mers for each latent dimension. These weights are obtained from the decoder parameters of BindVAE: $\theta \in \mathbb{R}^{M \times D}$, where $M$ is the size of the latent representation and $D$ is the number of k-mers in our input representation. Figure 1b shows this visualization where the x-axis has a total of 1068 k-mers and the y-axis shows the latent dimensions $i \in [0, 100]$. The set of 1068 k-mers was constructed by first selecting the top 20 8-mers for each latent dimension and then computing the union of the selected 8-mers. The $(i, j)^{th}$ entry of the matrix is $\theta_{ij}$. Based on the diagonal-heavy structure of the matrix, we can say that the model learns diverse patterns. The off-diagonal blocks show that similar k-mer importance patterns are learned for some of the dimensions. We find that these correspond to TFs from the same family.

Overall, we see two types of latent factors: ones that capture unique patterns such as dimension #0 and #72 (see enlarged inset below Fig. 1b) and ones that capture redundant patterns such as #69 and #83. The latent factors of the former type can be thought of as forming a basis, with each axis roughly corresponding to a different binding signature. Redundant dimensions are groups of dimensions that capture very similar k-mer distributions, i.e., have high weights for very similar k-mers, possibly due to binding sites of paralogous TFs.

### Motifs discovered de novo by our model

While Fig. 1b shows that diverse patterns are learned, we asked if these patterns represent coherent TF binding patterns. To answer this, we summarized the patterns learned by the model by doing a motif analysis for each latent dimension. Ideally, we would like to obtain motifs directly from the parameters of our model, i.e. using the distribution $\vec{\theta_m}$ for a latent factor $m$. However, the use of relatively short k-mers (8-mers) to represent the input, and the use of wildcards, limit the length and the accuracy of PWMs directly obtained from $\vec{\theta_m}$.

Given the caveats above, we adopted a postprocessing procedure that can use higher-order k-mers to get more accurate binding patterns. First, we generate the set of all possible 10-mer DNA sequences, $S^{10mers}$ (total of 524,800 10-mers without wildcards). The model trained on the ATAC-seq peaks is applied on $S^{10mers}$, where the input for each 10-mer example is the 8-mer counts vector of size $D$. The latent representation vector $\vec{z}_l$ for the $l^{th}$ 10-mer is its *latent score vector*. For a given latent dimension $k$, we can rank all the 10-mers based on the latent scores: $z_{lk}$ $\forall$ $l \in S^{10mers}$. Let the top 200 10-mer sequences in this ranking be $S^{200} = \{l_1, l_2...l_{200}\}$. We then construct a PWM from the DNA sequences in $S^{200}$ using MEME [15] and render it using SeqLogo [16] for consistent coloring. These results are tabulated in Fig. 1c for GM12878, with the third column showing the learned motifs. The second column shows the name of the TF assigned to the latent dimension by our procedure described in Algorithm 1 (discussed in the next section) and the fourth column shows the corresponding CIS-BP [17] motif. For brevity, we show the top few motifs in Fig. 1c, that were selected based on the *p*-values assigned by Algorithm 1.

We observe that the motifs learned by BindVAE are similar to the motifs from CIS-BP, which are based on in vitro and in vivo studies of individual TF binding. Since our input representation relies on 8-mers, the model is biased towards learning shorter motifs more accurately. We also find that TFs with long motifs are split across multiple dimensions, for instance RUNX3 is split across dimensions #60 and #39, and CTCF is split across dimensions #11 and #4.

Overall BindVAE finds motifs between 6 and 10 in width (average width 7.4). Our results illustrate that our VAE-based model learns binding preferences of representative TFs de novo.

### Mapping the TFs to dimensions

So far we showed that BindVAE is able to learn k-mer patterns of TF binding motifs. In order to validate that these learned motifs (Fig. 1c) are indeed meaningful to TF binding, we incorporated in vitro TF binding information derived from the HT-SELEX (high-throughput systematic evolution of ligands by exponential enrichment) study by Jolma et al. [18] In each TF experiment the HT-SELEX assay produces, at the end of several cycles of TF-bound oligonucleotide selection, oligomers that have a high affinity to bind the specific TF. The enriched probes thus represent TF binding preferences outside the cellular context. The HT-SELEX experiments in Jolma et al. [18] span hundreds of TFs from various TF families, out of which we use 296 TFs.

We use the oligomer probes that were enriched in each TF's HT-SELEX experiment to map the individual latent dimensions $i \in [1...M]$ from the bottleneck layer of BindVAE to TFs thereby making them amenable to biological understanding and facilitating further analyses. We project enriched probes into the latent space generated by the ATAC-seq peaks; i.e. we do inference on the probes using the BindVAE model trained on GM12878 peaks. Next, we use the procedure outlined in Algorithm 1 (please refer to the "Methods" section), where we compare the set of latent score vectors $\{z\}$ of one TF's HT-SELEX probes to those of the probes from all other TFs. A TF $t$ is mapped to a dimension $m$, generating the mapping $m \rightarrow t$, if $m$ ranks $t$'s probes higher than the probes from all other TFs.

We find that our algorithm produces all types of mappings between TFs and dimensions: one-to-one, one-to-many, many-to-one, and many-to-many. Homologous proteins or multiple members of a subfamily are mapped to the same dimension due to the similarity of their binding preferences. For instance, in the GM12878 model, the T-box family of TFs are assigned to the same latent dimension. Some TFs might be assigned to several dimensions for two reasons: (1) they have a long motif or (2) some homolog of that TF appears in the data but does not exist in our set of 296 TFs, which is limited by the HT-SELEX experiments. Another consequence of this limitation is that some latent dimensions are not assigned to any TF even if there is an enrichment in the DNA sequence pattern that they capture.

### Mapping peaks to TFs

We next 'assign' each ATAC-seq peak in the input data to the top 3 TFs for downstream analysis and visualization. Given the $M$-dimensional latent score vector $z_i$ of the $i^{th}$ peak, and the mapping $\mathbf{F}$ from dimensions to TFs, the assigned TFs are given by: $\mathbf{F}(\arg\max_{d \in [1 \cdots M]} z_{id})$, i.e., the 3 TFs corresponding to the highest latent scores. In general, we find that each peak's representation is spread across multiple latent dimensions. For example, a 30bp region of a peak from our GM12878 dataset in Fig. 2a contains TFs from two different families: ETS1 and another unmapped TF from latent factor #93.

### Latent factors capturing non-TF related patterns

In addition to TF-specific patterns being learned, some latent dimensions capture information pertaining to low complexity regions, genomic background, or Tn5 transposase cleavage bias. In Additional file 1: Fig. S1 we show how the top k-mers from latent factor #37 encode genomic background regions with GC-rich patterns, and in Additional file 1: Fig. S2 we show peaks with low complexity regions being assigned to the same topic. In Additional file 1: Fig. S3 we show peaks of two types, demonstrating the disentanglement done by BindVAE on peaks with low-complexity regions and TFBS. Further details of these experiments are in Additional file 1: Section 1.

### Projecting HT-SELEX probes into the latent space

From the previous sections we can conclude that BindVAE learns diverse and coherent patterns that map to TF binding preferences. In the following sections we explore what this entails for input DNA sequences of various lengths and types.

Just as in the previous section, we project the HT-SELEX probes into the latent space generated by the ATAC-seq peaks and visualize the results. Since HT-SELEX experiments use short probes that are 20bp long and capture in vitro binding affinities, they have very distinct patterns across different TFs. The heatmap in Fig. 2b shows the latent space for 10,000 probes, one probe per row, with the rows grouped by the TF that each probe was enriched for. There are 200 enriched probes (and hence rows) per TF experiment. We show 42 dimensions, out of the 100-dimensional latent space ($M = 100$), chosen based on whether a dimension contains any signal over the 10,000 probes. We see that a block diagonal structure emerges in this matrix, which indicates that several latent dimensions are orthogonal to each other and show distinct binding patterns. Note that each red square block corresponds to 200 probes.

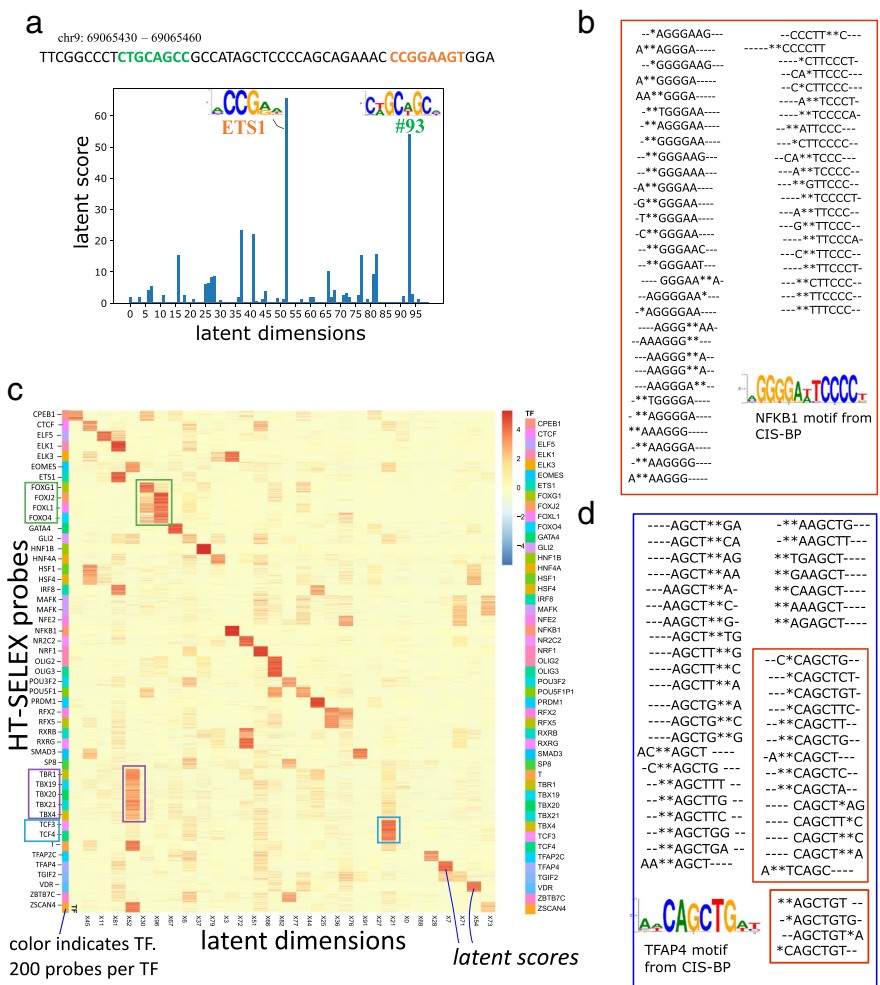

**Fig. 2 a** TFs and the corresponding 'latent scores', assigned to a 30bp region of a random peak from our GM12878 dataset. **b** Heatmap of the latent space obtained by our model upon doing inference on ≈10,000 SELEX probes from 48 TF experiments. Each row is a SELEX probe, with the probes being colored by their TF. There are 200 probes per SELEX experiment. The green boxes highlight the FOX family of TFs and are showing the corresponding probes' latent scores in dimension 96 (X96). **c**, **d** The top fifty 8-mers from two k-mer distributions learned by BindVAE are shown. These were obtained by sorting the decoder parameters encoding the k-mer distributions, namely $\theta_m \in \mathbb{R}^{112800}$ for the following two topics/ dimensions $m = 72$ and $m = 7$. The 8-mers have been aligned using multiple sequence alignment and the ∗ symbols show the wildcards from our k-mer representation. In **d**, the 8-mers in the two red boxes correspond to roughly the prefix (top box) and suffix (bottom box) of the motif

Further, TFs from a family share the same latent space due to the similarity in their DNA binding sites. For instance dimension #67 shows signal for several forkhead box (FOX) proteins: FOXJ2, FOXL1, FOXO4. Similarly, dimension #30 contains DNA binding preferences for T-box TFs TBR1, TBX19, TBX20, TBX21, and EOMES. In addition to T-box proteins, we see that dimension #30 also captures binding preferences for ZSCAN4, which is a C2H2 zinc finger. Dimension #21 encodes bHLH (basic helix-loop-helix) binding preferences for TCF3 and TCF4. In order to further understand the model's grouping of similar TFs into a single latent factor, we compare their CIS-BP motifs in Additional file 1: Fig. S4. We observe that the TF binding PWMs are very similar for each group and hence binding sites for these TFs get projected into the same region of the latent space by our model.

### Top 8-mers learned for NFKB1 and TFAP4

We illustrate the quality of the learned posterior distribution $p_\theta$, which is defined by the encoder parameters $\theta \in \mathbb{R}^{M \times D}$, by sorting the decoder parameters $\theta_m$ for each latent dimension $m$ and showing the top 50 corresponding 8-mers. Figure 2c shows the 8-mers from dimension $m = 72$ which is mapped to NFKB1 by Algorithm 1. For ease of interpretation, we align the 50 8-mers using Clustal Omega [19] and also show the CIS-BP motif corresponding to the mapped TF. NFKB1 is a palindromic motif of length ~13 and TFAP4 has a motif of length 10. For TFAP4, we see that the top 8-mers learned by our model can be partitioned into two groups based on their distinct patterns, with one group matching the beginning of the motif (pattern TCAGC) and the other matching the end (AGCTGT), as shown in Fig. 2d. In general, we observe that a single latent factor can capture the binding signature of TFs with motif length $\leq 10$. TFs with longer dimer motifs are also captured by a single latent factor, for instance NFKB1 in Fig. 2c and VDR in Additional file 1: Fig. S5. When the TF motif is longer than 10, we typically find the binding signature to be split across two latent factors – with one capturing the beginning and the other capturing the end of the motif, as seen for CTCF in Additional file 1: Fig. S5.

### Cell-type specific TF patterns

We find cell-type specific differences in the TFs whose binding signals are recovered by BindVAE, consistent with the different TFs expressed in these cell types (see Fig. 3d). There is some evidence that we also find differences in motifs for the same TFs between two cell lines, in cases where these TFs are expressed in both. We show some examples in Additional file 1: Table S2 for the two cell lines GM12878 and A549. We also compare all the k-mer distributions learned for both cell types by visualizing them together in the PCA plot shown in Additional file 1: Fig. S6. We apply PCA to the decoder parameters $\theta \in \mathbb{R}^{100 \times 112800}$, where each row is one k-mer distribution.

### Cooperative binding signals in GM12878

We find that multiple latent dimensions show k-mer patterns from diverse TF families, possibly indicating the presence of cooperative binding sites in the ATAC-seq peaks. Since the co-occurrence of multiple binding patterns might merely suggest that binding sites of two different TFs are present in a peak and not necessarily binding cooperatively, we use CAP-SELEX data [20] to validate this hypothesis. Jolma et al. [20] developed CAP-SELEX, an in vitro assay for studying interactions between pairs of DNA-bound TFs, using DNA sequences of length 40bp.

We analyze two latent dimensions: #60 in Fig. 3a and #67 in Fig. 3b, which show cooperative binding signals for MYBL1-MAX and FOXJ3-TBX21 respectively. In order to show the presence of cooperative binding patterns, we compare the latent scores of enriched CAP-SELEX probes from *pairwise* TF experiments to the latent scores of enriched HT-SELEX probes from *individual* TF experiments. We show the distribution of the latent scores along the y-axis, and the x-axis shows the source TF (or TF pair) experiment for each distribution. We also show the distribution of latent scores for probes from all other TFs, which shows that the pairwise signals are significantly higher than average. This is also indicated by the *p*-value (shown in brackets below each TF label) assigned by Algorithm 1. For example, the *p*-value of mapping dimension #67 to the TF pair FOXJ3-TBX21 is $2e^{-62}$, while that for all 'other TFs' is 1.0.

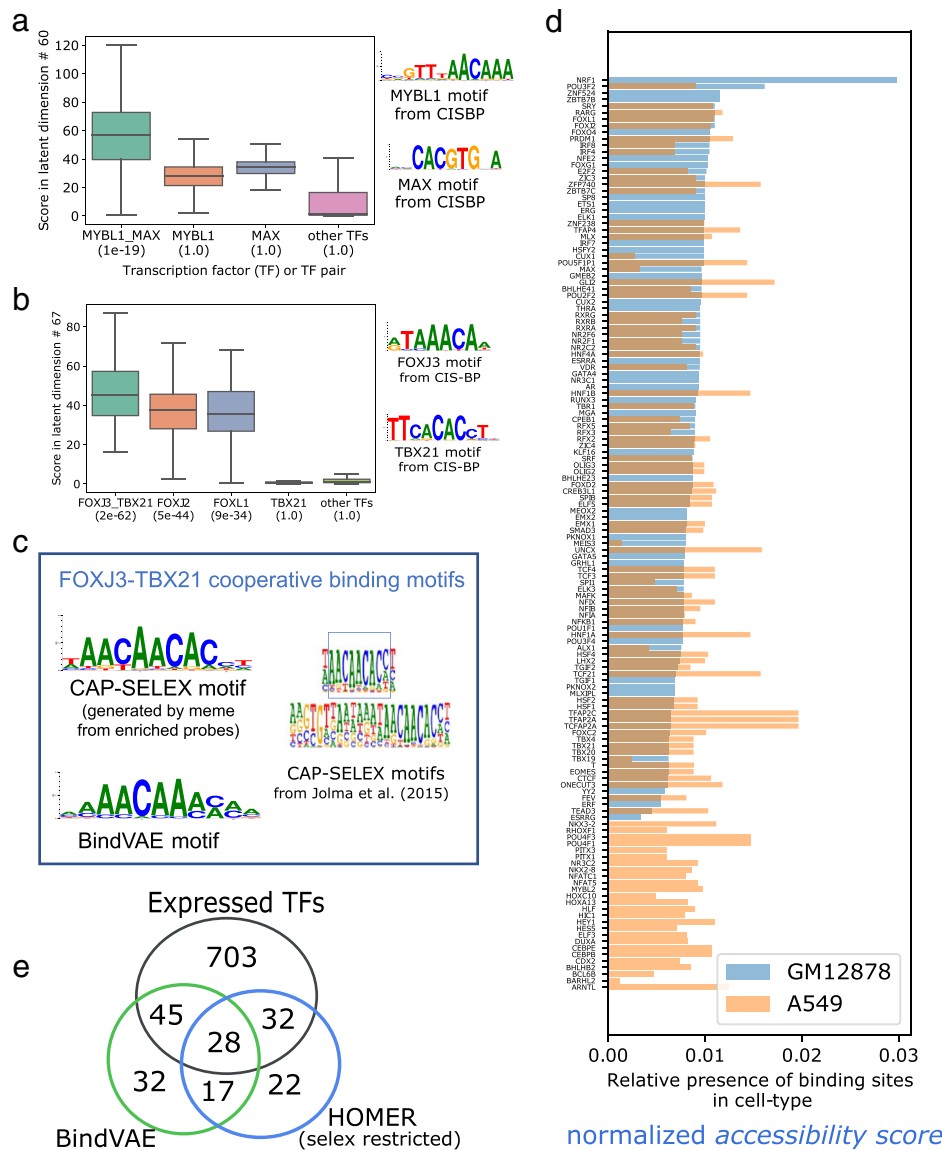

**Fig. 3 a**, **b** Analysis of cooperative binding signals learned by the model trained on GM12878 peaks by scoring HT-SELEX probes from individual TFs and CAP-SELEX probes from pairs of TFs. The y-axis shows the distribution of latent scores of individual probes for the TF experiments shown on the x-axis. In the brackets below each TF label, we show the *p*-value of assigning to the latent dimension, that particular TF, using Algorithm 1. Cooperative binding motifs can be enriched while the individual TF motif may not be enriched, as seen for MYBL1, MAX in **a** and TBX21 in **b** that have a *p*-value of 1. **c** *left:* CAP-SELEX based motif for the TF pair binding from Jolma et al. [20] and *right:* Motif learned by BindVAE for FOXJ3:TBX21 cooperative binding **d** TFs found in each cell type by our model and their 'accessibility scores' obtained by summing over all ATAC-seq peaks showing the possible extent of accessibility in the two cell lines: GM12878 (light blue) and A549 (orange). See text for details. **e** Venn diagram showing the overlap in the TFs found de novo by the VAE and HOMER in GM12878 and the extent of their overlap with all expressed genes. We show HOMER results run with the set of 296 motifs from Jolma et al. [18] (selex restricted) where HOMER only uses the TFs from our HT-SELEX set of 296 TFs

We further analyze the motif captured by dimension #67 for the TF pair FOXJ3-TBX21 in Fig. 3c by constructing a PWM from 15-mers ranked high by this dimension. Since the number of all possible 15-mers is prohibitively large, we sample 10% of them and run inference on these. We show the motif obtained by this process on the right side of Fig. 3c. On the left, we show the motif published by Jolma et al. [20] (using their algorithm)

for `FOXJ3-TBX21` cooperative binding. We show the motif learned for `MYBL1-MAX` in Additional file 1: Table S3, where we also show the 13 other TF pair motifs found by Bind-VAE. Note that combinations of TFs with a successful CAP-SELEX experiment are limited after filtering for quality metrics as described in BindSpace [7]. Composite motifs that differ from the concatenation of individual motifs are found for $\approx 70$ pairs of TFs, for which we can assess enrichment in the latent dimensions. Given the sparsity of CAP-SELEX data, we are unable to present a more comprehensive analysis of cooperative binding.

We further examine the results to understand if there is any intrinsic source of bias in CAP-SELEX probes or motif patterns. All examples of cooperative motifs that we find involve TFs with distinct binding motifs, i.e. the same nucleotide pattern is not repeated for each TF in the pair, which could confound cooperative motif analysis. Next, in Fig. 3a and Additional file 1: Fig. S7 (top) we see that the cooperative binding motif is enriched, while the individual TF motifs are not. Conversely, in Additional file 1: Fig. S7 (bottom) we show an example where the individual motif is enriched, while the cooperative motif is not. This shows that CAP-SELEX probes do not intrinsically achieve higher latent scores on account of either of the individual motifs being present.

### Accessibility patterns in different and similar cell types

In Fig. 3d we show the proportion of binding sites that we predict across various TFs in the two cell types in our study. Each bar represents one TF (or latent dimension *m*) and the height of the bar, which we call the '*accessibility score*' is obtained by summing the latent scores (or topic proportions) $z_{im}$ for all peaks $i$: $\sum_{i \in \text{all peaks}} z_{im}$ for each topic *m*. We only plot the latent dimensions *m* that were successfully mapped to TFs using Algorithm 1. The bars are colored by cell type, blue for GM12878 and orange for A549, overlapping areas appearing in grey. The bars are sorted in increasing order with respect to accessibility scores of TFs from GM12878. We see different accessibility patterns in these two cell types, and Gene Ontology (GO) term enrichment using PANTHER of the distinct TFs found for each cell type shows biological processes that are specific to each. For instance, we find 'myoblast fate commitment' and 'calcineurin-NFAT signaling cascade' being enriched in the TFs found for A549, whereas 'regulation of CD8-positive, alpha-beta T cell differentiation' and 'intracellular steroid hormone receptor signaling pathway' are enriched for GM12878.

We also illustrate the difference in activation patterns for *similar* cell types by training separate models on samples coming from the same cell type, but different donors: one male donor and one female donor for naive B cells. We plot the accessibility scores for TFs found by both models (Additional file 1: Fig. S8) and find that for similar cell types or replicates, there is overlap in the TF patterns.

Since BindVAE is a probabilistic model that optimizes the likelihood of the data, there is some variance/uncertainty across experimental runs. This is unlike autoencoders since the latent vector here is *sampled* from the Dirichlet posterior distribution. In addition, there is the variance from the randomized initialization of model parameters. We quantify this uncertainty by training models 5 times using identical hyperparameters, on data from only the female donor sample of naive B cells. We plot the accessibility scores for the union of all TFs found, along with the standard deviation across the 5 repeat runs. While there is variance in the exact TFs found across the repeat runs, if we visualize the k-mer distributions for two of the replicate experiments in Additional file 1: Fig. S6 (top right),

we see that they are quite similar. Compare this to the k-mer distributions learned for two distinct cell types in Additional file 1: Fig. S6 (bottom left) or for the same cell-type but samples coming from different donors in Additional file 1: Fig. S6 (top left).

### Comparison with de novo motif discovery methods

We compare BindVAE with three other motif discovery approaches: HOMER [21], MEME [15], and GADEM [22]. The last two only produce motifs and are hence run in conjunction with a motif-matching approach, Tomtom [23] to find the matching TFs. We find that BindVAE has a higher precision but often worse recall and F1 score (Additional file 1: Table S4a, S4b), while BindVAE+Tomtom, where we replace Algorithm 1 with Tomtom, has a higher recall and F1. We find that Algorithm 1 trades off precision for lower recall (see details in Additional file 1: Section 2, where we also compare run-times of the approaches). We find that BindVAE is one of the faster methods, particularly since the prior motif discovery approaches do not scale well to large-scale NGS datasets and the efficiency is dependent on the e-value/$p$-value threshold used. Here, we briefly present our observations with HOMER on the GM12878 dataset.

HOMER [21] is an unsupervised motif discovery algorithm that uses differential enrichment to find motifs. It compares peaks to background DNA sequences and tries to identify patterns with mismatches and gaps that are specifically enriched in the peaks relative to the background. We compare our unsupervised VAE-based method to HOMER on de novo motif detection in GM12878. Figure 3e shows the overlap in the TFs found by both approaches and how many of these are expressed in GM12878. Since the TFs detected by BindVAE are restricted by our post-processing algorithm that relies on HT-SELEX data, we present the HOMER results in a similar configuration for a fair comparison: 'SELEX restricted', where we use the HT-SELEX PWMs published by Jolma et al. [18] for the 296 TFs that we consider.

We find that HOMER finds a total of 99 motifs that are mapped to known TFs within our HT-SELEX set of 296 TFs. BindVAE finds 122 motifs from latent dimensions mapped by Algorithm 1. Looking at expressed TFs, 60/99 ($\approx$ 60%) found by HOMER and 73/122 ($\approx$ 60%) found by BindVAE are expressed. Therefore HOMER finds fewer TFs overall than BindVAE, and a similar proportion are expressed in GM12878 for both methods. We provide a table of the learned and expressed TFs in the Additional file 1: Table S5. We also present a detailed comparison with HOMER for varying $p$-value and e-value cut-offs in Additional file 1: Table S4b.

### Projecting ChIP-seq peaks into the latent space

In Fig. 4a,b, we show a two-dimensional projection using Uniform Manifold Approximation and Projection (UMAP) [24] of the learned representations for ChIP-seq peaks from GM12878 for three transcription factors with distinct binding preferences. There are 1000 peaks from each ChIP-seq experiment, and each peak is colored by the TF. Figure 4a shows proteins from different families: CTCF, a C2H2 zinc finger, MAFK a bZIP TF, and ELK1 that contains an ETS domain. Here we see that our model learns distinct embeddings for TFs that have different binding preferences. However, embeddings of CTCF peaks are more distributed due to the ability of its zinc finger domains to bind to heterogenous DNA sequences. Figure 4b shows FOS a bZIP TF, IRF4 an IRF family transcription factor and SPI1 that contains an ETS domain. We find that embeddings of

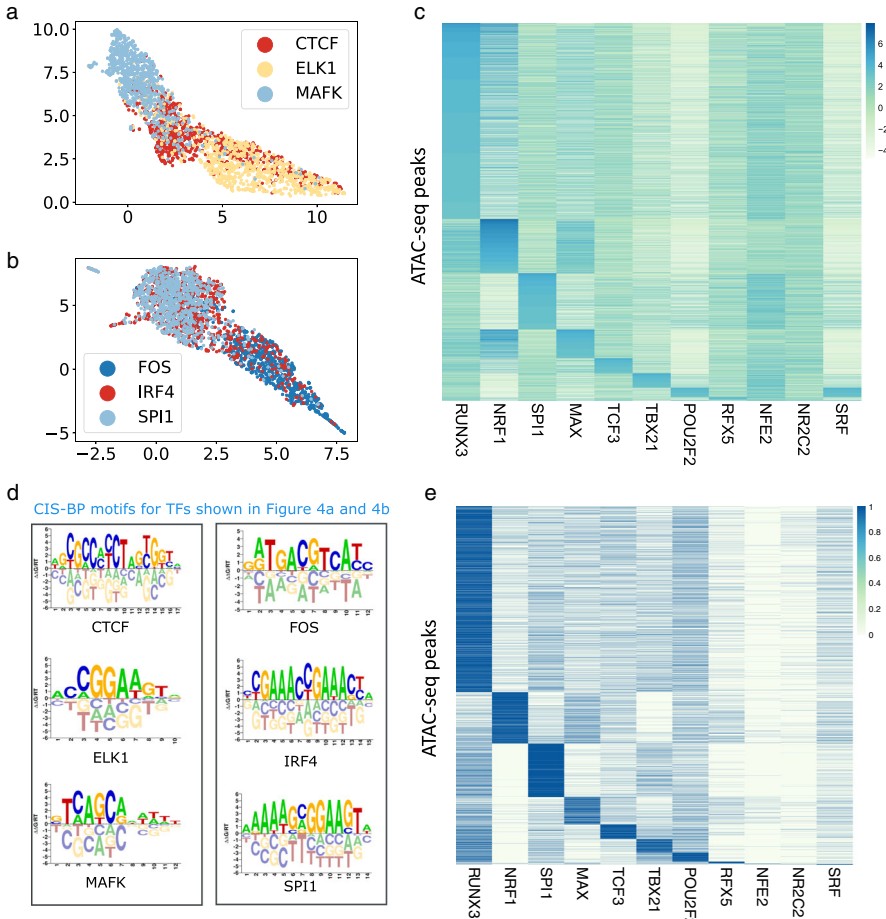

**Fig. 4 a** UMAP projection of the latent representation for ChIP-seq peaks from `CTCF`, `ELK1` and `MAFK` **b** UMAP projection of the latent representation for ChIP-seq peaks from `FOS`, `IRF4` and `SPI1`. For each TF, 1000 peaks are shown. **c** Heatmap showing 38,496 ATAC-seq peaks from GM12878 cell line, that overlap with ChIP-seq peaks from 11 different TFs. For a TF *t*, the peaks have been sorted by the latent score of the dimension that was mapped to *t* by Algorithm 1 **d** CIS-BP motifs for TFs shown in **a** and **b** show distinct motifs for all TFs. **e** Matrix of GM12878 ATAC-seq peaks ordered as per the matrix ranked by latent factor scores (topic scores) from Figure 4c. Rows are peaks and columns are TFs. Each entry *i, j* in the *i*-th row and *j*-th column of the matrix is the *i*-th peak's ChIP-seq annotation for TF *j*

peaks from `FOS` are distinctly clustered while those from `IRF4` and `SPI1` largely overlap despite a low overlap in the genomic regions of the peaks (34 out of 1000 peaks overlap). This is possibly because `IRF4` binds DNA weakly but cooperative binding with factors such as SPIB in B cells increases binding affinity [25]. We notice a similar pattern of overlap with `BATF` and `JUND` which are both bZIP TFs that form heterodimers while binding DNA [26].

To analyze whether our model learns meaningful representations, we use ChIP-seq as a source of ground truth and verify whether known binding sites for a given TF are transformed to the same latent dimension by our model. We look at the intersection of the TFs in our HT-SELEX set (296 TFs) and those for which we have reliable ChIP-seq data, which gives us 11 TFs. For each of these TFs, we find all ATAC-seq peaks that have an overlap of at least 50bp with any ChIP-seq peak and plot the latent dimension that was assigned to that TF. In Table 1 we show the number of overlapping peaks between the ChIP-seq experiment and our GM12878 dataset. We show the resultant matrix for these

**Table 1** Statistics of ChIP-seq data used to plot Figure 4c showing the number of overlapping peaks

| TF-name | Peaks overlapping with ATAC-seq data | Peaks from column 1 that are mapped to TF by Bind-VAE | Precision | Recall |
|---------|--------------------------------------|------------------------------------------------------|-----------|--------|
| TCF3 | 9703 | 5107 | 0.17 | 0.53 |
| MAFK | 782 | 436 | 0.15 | 0.56 |
| ELK1 | 4484 | 3773 | 0.15 | 0.84 |
| NRF1 | 4322 | 3874 | 0.21 | 0.90 |
| RFX5 | 3226 | 1506 | 0.11 | 0.47 |
| NR2C2 | 410 | 158 | 0.15 | 0.39 |
| TBX21 | 14820 | 5865 | 0.25 | 0.40 |
| POU2F2 | 14972 | 4593 | 0.16 | 0.31 |
| SPI1 | 16177 | 10496 | 0.33 | 0.65 |
| RUNX3 | 33873 | 25403 | 0.51 | 0.75 |
| MAX | 9110 | 6793 | 0.21 | 0.75 |
| ETS1 | 7285 | 5373 | 0.28 | 0.74 |

11 TFs by depicting it as a heatmap in Fig. 4c. The heatmap shows a centered log ratio (CLR) transform of the latent representations, with rows representing peaks and columns representing TFs. Our approach gives us a total of ≈38,496 peaks across the 11 TFs which we sort by their membership, with peaks that belong to RUNX3 being shown at the top of the heatmap as it has the largest number of mapped peaks.

We show the TF ChIP-seq experiment connected with each row of Fig. 4c, in a second heatmap in Additional file 1: Fig. S14, that indicates the ChIP-seq ground truth label assigned to the ATAC-seq peak via overlap (i.e. a minimum of 50bp overlap between a ChIP-seq and ATAC-seq peak to assign the TF label) in the same order as Fig. 4c. We see that some of the co-binding relationships found by BindVAE: such as NRF1 and MAX are explained by the overlap in ChIP-seq peaks.

## Discussion

Supervised deep learning methods for the prediction of TF occupancy data and chromatin accessibility are numerous, ranging from early deep convolutional neural network based models such as DeepSEA [10] and Basset [9] to more recent approaches usually mirroring advances in deep learning methods for natural language processing, such as the LSTM-based DanQ [27], Basenji using dilated CNNs [11], DeepSite [28], and DNA-BERT [29]. Naturally, it is possible to train discriminative deep learning models on sequence data to predict chromatin accessibility either as a binary label (open/closed) or in the regression setting, as many authors have done (e.g. the Basset and Basenji models [9, 11]). While these models have produced highly accurate predictions of TF occupancy, interpreting supervised models requires attribution of the learned parameters to the output labels—which is often not a robust process as many works in the broader deep learning literature have shown. For example, in medical image analysis, image features that are irrelevant for clinical analysis of the specimen can be used by the model to improve the prediction accuracy based objective function. In regulatory genomics, methods to interpret the sequence information captured by deep discriminative sequence models (e.g. DeepLIFT [30]) require detailed feature attribution over a large input window (such as 500bp to 1Mbp), and in general these methods do not generalize across cell types. Other approaches to improve the interpretability of these models (e.g. use of a Fourier prior in

training [31]) are still in their infancy. Interpretation of supervised deep sequence models has been more successful where models are trained on high-resolution TF occupancy data [12], since the underlying motif grammar is less complex than that for chromatin accessibility data.

Motif-matching methods such as FIMO [32] have been popular in biological studies due to the wide range of TFs, ease of application, and inherent unsupervised nature where the available PWMs can be applied on any DNA sequence. However, these approaches may return hundreds of motif hits for genomic regions of the size used in ATAC-seq analysis, i.e. 100bp to 200bp regions under peak summits.

Given the limitations of supervised models for interpreting chromatin accessibility, we explored an unsupervised deep learning model that learns binding patterns given open chromatin regions derived from ATAC-seq and is thus complementary to existing work. There are currently ≈1500 DNase-seq and ATAC-seq datasets spanning hundreds of cell and tissue types on the ENCODE portal, whereas datasets on TF binding experiments such as ChIP-seq are restricted to a few TFs per cell type, with the exception of a few highly profiled ENCODE cell lines. Hence, unsupervised or semi-supervised approaches may be desirable for decoding the TF binding landscape on less studied cell types.

Generative models, such as VAEs, are also more interpretable by design. For example, BindVAE seeks to capture individual TF binding signals as latent dimensions in the model, as well as disentangling other sequence signals like low-complexity k-mer signatures. More broadly, generative models learn universal representations of the data that can then be used in downstream task-specific applications. We propose that such methods are the next step in the evolution of machine learning modeling on genomics data since supervised models are typically not transferable to other prediction tasks. Further, unlike recent language-based deep learning models, our VAE based model is easily scalable for training on data from several cell types and can also be run in a multi-processor computational environment without GPU support, albeit at lower efficiency.

The input to BindVAE uses 8-mers with wildcards, which allows us to interpret the learned latent factors. There is a rich literature on the use of k-mers for representing DNA and protein sequences, from early works using oligonucleotide frequencies [33] and k-mer based string kernels using support vector machines [4, 34]. We rely on the results from these prior works that show the robustness of this representation.

While we lose the exact positional information due to our choice of bag-of-words as input, we do not lose any context information as the surrounding DNA sequences are still part of the input representation; for this broader sequence context, the exact position of a specific k-mer in the 200bp peak may not be important for a successful encoding. Secondly, the bag-of-words representation actually gives us flexibility – the input DNA sequence can be of arbitrary size, and we can apply the same model on inputs of varying size from diverse in vivo and in vitro data. This is not true for a one-hot-encoding based input representation used by other deep learning models, which requires the DNA sequence inputs to be of a fixed constant size. We find that there is a trade-off between the complexity of the input representation and the interpretability of the resulting model. Our choice of a bag-of-words input makes the model readily interpretable by giving us an easy way to map components of the latent space to distributions over k-mers. Lastly, it would be straightforward to add positional information in our model the way transformers do, by adding position encodings.

Our model also does not use the real-valued peak accessibility as an input and therefore does not capture the relative frequency of binding patterns that are in less accessible regions vs. more accessible regions. However, we note that many computational methods for finding TF binding signals in bulk ATAC-seq make the same simplifying assumption of treating reproducible peaks as positive examples—without retaining accessibility values—and perhaps using inaccessible regions or shuffled genomic sequences as a negative background. For example, virtually all PWM-based motif finders (HOMER, MEME) are blind to real-valued accessibility information, and even the widely-used discriminative model gkm-SVM trains with binary rather than real-valued labels. That said, one straightforward extension to include accessibility information in BindVAE's k-mer count based input would be to scale the counts based on accessibility of the region that each k-mer comes from.

We found that our VAE based model can learn distinct binding patterns from ATAC-seq peaks without any TF labels. Of the 102 distinct patterns learned over the latent dimensions, we found specific patterns for some TFs and were able to map the latent factors to unique TFs. In contrast, for others, we found a coarser pattern that corresponds to one of several TFs from a family, such as `T-box` proteins. Paralogous TFs are difficult to learn as separate factors due to the highly similar patterns in their binding sites, which cannot be captured uniquely by a distribution over 8-mers with wildcards in an unsupervised fashion. Our model also learns combinations of patterns for TFs that co-occur within peaks and that are involved in cooperative binding, and analyzing these patterns produced composite motifs. Using higher-order k-mers in our model can improve the coverage over longer motifs. However, this would increase the input dimension of BindVAE significantly and thereby increase computation overhead substantially.

## Methods

We begin this section by introducing some terminology and notation used throughout the paper. We use $\mathbf{z}$ to refer to the latent variable, $\mathbf{x}$ to refer to the input variable, $\vec{z}$ or $\vec{z}_i$ to refer to an instantiation of the latent variable, i.e. to a latent vector corresponding to an input $\vec{x}_i$, and $\vec{z}_{ik}$ to refer to a component of the latent vector. $M$ is the size of the latent space or the number of latent dimensions, i.e. the size of the bottleneck layer of the VAE. $D$ is the size of the input and – since our model is an autoencoder – also the size of the output space.

### Variational autoencoders

Variational autoencoders (VAEs) [35, 36] are latent variable models that combine ideas from approximate Bayesian inference (variational inference) and deep neural networks, resulting in a framework that can use backpropagation-based training.

Let $\mathbf{x}$ represent the data and $\mathbf{z}$ be the latent variable. VAEs express the joint distribution $p(\mathbf{x}, \mathbf{z}) = p(\mathbf{z})p(\mathbf{x}|\mathbf{z})$ where $p(\mathbf{z})$ is a prior distribution over $\mathbf{z}$, i.e $\mathbf{z} \sim p(\mathbf{z})$, and $p_\theta(\mathbf{x}|\mathbf{z})$ is the likelihood function. In the context of neural networks, $p_\theta(\mathbf{x}|\mathbf{z})$ is the probabilistic decoder that generates data $\mathbf{x}$ given latent variables $\mathbf{z}$, with the goal of reproducing $\hat{\mathbf{x}}$ that is close to $\mathbf{x}$. Since estimating the true posterior distribution $p_\theta(\mathbf{z}|\mathbf{x})$ is often intractable, an approximate posterior distribution (also known as the variational distribution) $q_\phi(\mathbf{z}|\mathbf{x})$ is used, which is formulated by the probabilistic encoder in the neural network model. The encoder outputs $\mathbf{z} \sim q_\phi(\mathbf{z}|\mathbf{x}) = q_\phi(\mathbf{z}|\eta)$ where $\eta = MLP(\mathbf{x})$ is computed from

the observation **x** by a multi-layer perceptron (MLP). Figure 1a shows a neural network depiction of this model.

**Loss function**: VAEs optimize the parameters $\phi$ and $\theta$ of the encoder and decoder jointly by maximizing the evidence lower bound (ELBO) using stochastic gradient descent. The ELBO is the variational lower bound on the marginal log-likelihood of the data $\log p_\theta(\mathbf{x})$ and is given by:

$$
\begin{aligned}
\log p_\theta(\mathbf{x}) &= \sum_{i=1}^{N} \log p_\theta(x_i) \\
&\geq \sum_{i=1}^{N} \mathbb{E}_{q_\phi(z_i|x_i)}[\log p_\theta(x_i|z_i)] - KL(q_\phi(z_i|x_i)\|p(z_i))
\end{aligned}
\tag{1}
$$

The first term is the reconstruction loss, i.e. the error in reconstructing $x$ using $z$, and the second term is the Kullback-Leibler (KL) divergence between the posterior distribution $q_\phi(\mathbf{z}|\mathbf{x})$ and the prior distribution $p(\mathbf{z})$. The likelihood term $\log p_\theta(x_i|z_i)$ tries to maximize the probability of reconstructing the input $x_i$ from $z_i$ and is formulated as a multinomial distribution, with the decoder parameters $\theta$ containing the probability vectors for each component of the distribution. The reconstruction term tries to improve the quality of the reconstruction without regard to the properties of the latent space, while the KL term acts like a regularizer and constrains the latent representations within the space imposed by the prior distribution.

Since our goal is to incorporate k-mer distributions in the model, we use a Dirichlet distribution as a prior on the latent variables instead of the more prevalent normal distribution used in Gaussian VAEs, which are difficult to interpret as the bottleneck layer $z$ and can take arbitrary values. On the other hand, a Dirichlet distribution will only allow non-negative latent variables; therefore the value taken by each latent dimension $m$ given a particular **x** can be considered a 'membership', with larger values indicating stronger membership.

However, VAEs with a Dirichlet prior cannot be trained using the explicit reparameterization trick [35], where a Gaussian variable $\mathbf{z} \sim N(\mu, \sigma^2)$ is reparameterized as $\mathbf{z} = \mu + \epsilon\sigma$ with $\epsilon \sim N(0, 1)$, thus allowing the gradient to be backpropagated through the latent variable $z$. This is because no such simple variable transformation is possible for the Dirichlet distribution. We thus use the implicit reparameterization gradients-based approach developed by Figurnov et al. [37] which provides unbiased estimators for continuous distributions that have numerically tractable cumulative distribution functions (CDFs). To incorporate a Dirichlet distribution, they use the property that it can be rewritten as a composition of several univariate Gamma variables. We refer interested readers to Table 1 from Figurnov et al. [37] for the equations that show the computation of implicit gradients for backward propagation through a node with a Gamma distribution.

**Layers:** Our encoder has 3 fully connected layers with 300 hidden units in each layer. The decoder simply maps from the bottleneck layer to the output reconstruction layer via $p_\theta$.

**Latent score vector:** We let $\vec{z}_i \in \mathbb{R}^M$ denote the latent representation vector for the $i^{th}$ input DNA sequence, obtained upon inference via the BindVAE model as its latent score vector. We will often refer to latent scores w.r.t certain latent dimension $k$, which will simply be the value $\vec{z}_{ik}$.

### Vocabulary or input space

Unlike several other deep learning models that use a one-hot encoding of the raw DNA sequence, we use k-mer features to capture sequence preferences. We use a window of 200 bp around the peak summit and assume that the TF binding site can be present at any location in this window. Inspired by prior work [6, 7] and the wildcard kernel [38], we use all k-mers of length 8 with up to two consecutive wildcards allowed per k-mer to define the input space. We consider exact-matching k-mers and k-mers with wildcards as distinct features: for example, TATTACGT, TANTACGT, TNNTACGT are all counted separately. Further, an 8-mer and its reverse complement are treated as a single feature that combines the counts of both the 8-mer and its reverse complement. This results in a vocabulary or input space of size $D = 112800$.

### Parameter tuning and model selection

The hyperparameters of our model are the following: the dimension $M$ of the latent space/bottleneck layer, the number of layers and the width for the MLP of the encoder, the Dirichlet prior hyperparameter $\alpha$ that controls the prior distribution $\vec{\alpha}$ of topics, and the vocabulary size for the k-mer representation. We tried the following values for $M$: 10, 50, 100, 200, 500, 1000. For $\alpha$ we tried: $1e^{-3}, 1e^{-2}, 0.1, 1, 10, 20, 30, 50, 100$. Note that $\vec{\alpha} = \alpha \mathbf{1}^{1 \times M}$.

We found that increasing $\alpha$, which controls the prior of the Dirichlet distribution, increases the extent of overlap between the basis vectors defining the latent space (i.e. more sharing between the topics). In the extreme, this can lead to the so-called 'averaging affect' that variational autoencoders are known to suffer from, where the model learns an 'average' representation of the data. Further, very large values such as $30, 50, 100$ lead to convergence issues during optimization since the non-negativity constraints on $\theta$ are not met. Small values of $\alpha$ such as $1e^{-2}, 1e^{-3}$, due to the nature of the Dirichlet distribution, lead to a peaky prior distribution that tries to enforce each peak to have only one 'active' latent dimension. However, this leads to a lower likelihood as it does not capture the heterogeneous nature of peaks. We find that $\alpha \in [10, 20]$ results in models with a good trade-off between the diversity of the posterior and the likelihood (i.e. the loss function). We also find that as $\alpha$ changes, the learned topic distributions vary and result in different TFs being learned based on the prior. We keep $\alpha$ fixed for the initial several epochs (we set a burn-in of 150,000 steps, which is also a tunable parameter) and then optimize over $\vec{\alpha}$ by backpropagating the corresponding gradients.

Increasing the dimension of the latent space from $M = 10$, as expected, leads to the bottleneck layer learning more diverse patterns up to $M = 100$. For higher values such as $M = 200, 500, 1000$ the redundancy across dimensions increases substantially, i.e. several $\theta_i$s will be similar to each other. We tried various batch sizes and found 128 to be optimal. We used the Adam optimizer with a learning rate of $3e^{-4}$ and terminating optimization upon a maximum of 300,000 steps.

We select the final model based on the number of TFs mapped, i.e. the number of TFs that satisfy the $p$-value threshold of 0.05, in the procedure outlined in Algorithm 1. Given our observation about complementary sets of TFs being learned as we change the prior through $\alpha$, we decided to use an ensemble model. We pick the three best models, where we rank the models based on the number of learned TFs, and aggregate the non-redundant TFs from them to get the set of all learned meaningful dimensions.

**Model training time**: All experiments were run on Microsoft Azure Virtual Machines. With a single GPU, our code takes 4 to 5 hours to train for 300,000 epochs. Inference on 50,000 test examples takes $\approx 1$ minute.

### Mapping latent dimensions to TFs

We describe the algorithm used for mapping latent dimensions to TFs. If the probes of TF $t$ are ranked higher than all other TFs' probes by latent dimension $m$, the label $t$ is assigned to dimension $m$. The significance of this "enrichment" of $t$'s probes is computed by the Mann-Whitney U test. Note that this procedure can lead to a many-to-many mapping between dimensions $m \in \{1 \ldots M\}$ and TFs $t \in T$.

---

**Algorithm 1:** Map latent dimensions to TFs

---

**Input**: $M \leftarrow$ number of latent dimensions

$\{1 \ldots M\} \leftarrow$ set of latent dimensions

$N = 65,000$ (number of HT-SELEX probes)

$T$: the set of 296 TFs

$\vec{y} \in \mathbb{R}^N$: labels vector, where $y_i = t$ if probe $i$ comes from TF $t$

$\mathbf{z} \in \mathbb{R}^{N \times M}$: latent representation of all HT-SELEX probes

**Output**: Mapping $F : \{1 \ldots M\} \rightarrow T$

1 **foreach** $t \in T$ **do**

2 $\mathbf{z_t} \leftarrow$ rows of $\mathbf{z}$ for which $y_i = t$

3 $\mathbf{z_{-t}} \leftarrow$ remaining rows of $\mathbf{z}$, for which $y_i \neq t$

4 **foreach** $m \in \{1 \ldots M\}$ **do**

5 Let $a_{.m}$ be the $m^{\text{th}}$ column vector of $\mathbf{z_t}$

6 Let $b_{.m}$ be the $m^{\text{th}}$ column vector of $\mathbf{z_{-t}}$

7 $p_{tm} \leftarrow$ $p$-value of Mann-Whitney U test ($a_{.m} >$ top 5% of $b_{.m}$)

8 **if** $p_{tm} < 0.05$ **then**

9 $F \cup (m, t)$

 **end**

 **end**

**end**

---

### Datasets

#### ATAC-seq data from cell lines

We downloaded the publicly available GM12878 ATAC-seq dataset from the GEO database (accession GSE47753[1]). We took only samples generated using 50,000 cells. Since replicate 1 has much higher sequencing depth than the other replicates, we combined replicates 2, 3, and 4 to obtain a second replicate (renamed replicate 2). We followed the ENCODE ATAC-seq processing pipeline (https://www.encodeproject.org/atac-seq/) for pre-processing. Raw fastq files were adapter-trimmed using Trimmomatic and aligned to hg19 genome using Bowtie2 with default settings. PCR duplicates were then removed using Picard MarkDuplicates and Tn5 shifts are adjusted for. Peak calling was performed for each replicate using macs2 with parameters: `-nomodel -shift -37 -extsize`

---

[1]https://www.ncbi.nlm.nih.gov/geo/query/acc.cgi?acc=GSE47753

73. Finally, IDR was performed with the idr package and reproducible peaks were called with an IDR cutoff of 0.05. We identified a total of 76,218 reproducible peaks in the GM12878 ATAC-seq dataset. We downloaded the publicly available A549 ATAC-seq dataset from the ENCODE portal (accession ENCFF548PSN[2]). The details of the three other datasets used are in Additional file 1: Table S1.

### HT-SELEX

HT-SELEX (High-Throughput Systematic Evolution of Ligands by EXponential enrichment) is an in vitro experimental protocol that involves an iterative procedure that starts with an initial library of random oligonucleotides (oligos) of fixed length of either 20 or 40bp. Since this binding happens outside a cellular environment, it represents a TF's intrinsic DNA-binding preferences. At every iteration of the procedure, the input pool of oligos compete to bind to the TF. Oligos that do not bind at all or bind weakly are washed out from the pool while the rest are amplified using PCR. A sample of the amplified pool is sequenced to allow for computational analysis while the rest of the pool is used as input for the subsequent selection round. In this way, at the end of each round there are more high-affinity oligos in the pool than before, while non-binders and weaker binders are gradually eliminated.

We used the filtered HT-SELEX probes from Yuan et al. [7] for training our models. Briefly, this dataset contains HT-SELEX data sequenced in Jolma et al. [18] (ENA accession ERP001824) and in Yang et al. [39] (ENA accession ERP016411), which together constitute 547 experiments for 461 human or mouse TFs. The experiments that were filtered out were those showing: 1) poor consistency of 8-mer enrichment in consecutive HT-SELEX cycles, 2) low number of enriched probes, or 3) low diversity of probe enrichment. For each remaining experiment, the top 2,000 enriched 20-bp probes were selected per experiment. The filtered dataset contains 325 high quality experiments covering 296 TFs.

### ChIP-seq data: GM12878

Conservative and optimal IDR (irreproducible discovery rate) thresholded 'narrowpeak' files of ENCODE ChIP–seq data were downloaded from the ENCODE portal[3] for GM12878. We only downloaded ChIP-seq datasets which had a green 'Audit category' at the time of writing. We excluded experiments with 'Audit category' = orange or red as these have insufficient read length, insufficient read depth, poor library complexity, or partially characterized antibody. This gave us 82 experiments.

### CAP-SELEX

Consecutive affinity-purification systematic evolution of ligands by exponential enrichment (CAP-SELEX) [20] is an approach to identify TF pairs that bind cooperatively to DNA. It is again an in vitro assay based on a consecutive affinity-purification protocol coupled with enrichment of bound ligands. For a given pair of TFs, say $TF_1$ and $TF_2$, we downloaded the probes from cycle 4 and selected candidate probes for cooperative binding by picking frequent probes where the PWM model for the pair $TF_1$ and $TF_2$ (see the supplementary material from Jolma et al. [20]) was found. We used the MAST algorithm [40] for motif matching with a high e-value cut-off of 10, due to the relatively short length

---

[2] https://www.encodeproject.org/files/ENCFF548PSN/
[3] https://www.encodeproject.org/

of 40bp of the probes. We selected the top 1000 probes (or fewer, as found) and ran inference on them to obtain their latent representations. Further details about this dataset can be found in the Additional file 1.

### *RNA-seq expression data*

For GM12878, we downloaded gene expression data from the ENCODE portal with accession numbers: ENCFF906LSJ and ENCFF630BDD. We consider a gene to be expressed if it has an average expression of 0.05 RPKM or higher over the two datasets. For A549 we downloaded ENCFF203NNS, and for the T cell female donor sample we downloaded ENCSR336VTK[4]; we used a RPKM/FPKM cut-off of 0.05 to decide whether a gene is expressed or not.

### HOMER motif analysis

We first processed the set of HT-SELEX PWMs from Jolma et al. into a database format that is used as input. HOMER was then run in the de novo motif discovery mode by invoking the perl script findMotifsGenome.pl in the following manner:

```
./findMotifsGenome.pl peak-file.bed hg19/hg38 output-dir -S
1000 -p 10 -size given -len 6,8,10,12 -noknown -mset jolma -e
0.1
```

### Supplementary Information

---

Additional file 1. This is the supplementary file providing additional results, figures and tables that are referenced in the paper.

- **Table S1:** Details of the ATAC-seq datasets used in the analyses shown in the supplementary material.
- **Figure S1:** The top 100 8-mers from the latent dimension #37, which captures genomic background in the GM12878 model.
- **Figure S2:** The latent representation learned for top low complexity regions (LCRs) are similar as shown in the heatmap, where most of them have a high value for the same latent factor (#37).
- **Figure S3:** Latent projection of peaks with only repeat regions and peaks with some low-complexity repeating patterns and TFBS for a TF. This shows the disentanglement achieved by BindVAE.
- **Figure S4:** CIS-BP motifs for TFs from the same family or for paralogous TFs are shown, to illustrate the difficulty of learning TF-specific patterns for these. We show the TFs from the heatmap of Figure 2b (TFs in the boxes). Each group of TFs gets projected to the same latent factor by our model as discussed in the main paper.
- **Figure S5:** GM12878: Top 50 8-mers from some latent dimensions, aligned using Clustal Omega to summarize the patterns found. The CIS-BP motif corresponding to the TF that was assigned to each latent dimension (using Algorithm 1) is also shown. Since CTCF is assigned to multiple latent dimensions, the top 25 8-mers from each are shown.
- **Figure S6:** PCA performed on the decoder parameters $\theta \in \mathbb{R}^{M \times D}$ that capture the k-mer distributions, from models trained on various cell-types. Each dot on a plot represents a TF binding pattern or k-mer distribution. **(Top left)** k-mer distributions from two models trained on naive B cells from two human donors, male and female. **(Top right)** k-mer distributions from two models from two repeat experiments on female naive B cells. In both cases, there is not much variance in the learned patterns, as these are biologically close samples. **(Bottom left)** k-mer distributions from the two models trained on GM12878 and A549. Since these are distinct cell types, we see two distinct manifolds in the patterns being learned. There is also a cluster of latent dimensions at the bottom of the plot that captures similar k-mer patterns. **(Bottom right)** k-mer distributions from four models: GM12878, A549, mouse CD8 T cells, and human CD8 T cells.
- **Table S2:** Motifs constructed from latent factors learned for GM12878 and A549 for TFs learned by our model. HNF4A, NFIA, SRY have the same 'accessibility score' in both GM12878 and A549 trained models (see Figure 3d in the main paper). ELF5 and OLIG3 are both expressed in both cell types.
- **Table S3:** MYBL1:MAX motifs computed from the GM12878 model are shown in the top row. The bottom table shows all cooperative binding pairs of TFs found for GM12878 by our model using the CAP-SELEX data are shown.

---

[4]https://www.encodeproject.org/experiments/ENCSR336VTK/

- **Figure S7:** Examples of CAP-SELEX probes scoring higher and lower than individual TF probes. **(Top)** Example of cooperative binding CAP-SELEX probes of TFAP4:FLI1 being enriched while the individual TF probes are not enriched for TFAP4 or FLI1 ($p = 1$). **(Bottom)** Example where individual TF probes from the SELEX experiment for EOMES are enriched, while CAP-SELEX probes of cooperative binding between MYBL1:EOMES are not.
- **Figure S8:** Accessibility scores of TFs obtained by summing the latent representations over all ATAC-seq peaks showing the possible extent of accessibility for each TF. The datasets used here are described in Table S1. **(Left)** TFs found in naive B cells by our model and their 'accessibility scores' in the two human donors: male donor (light blue) and female donor (orange).**(Right)** Averaged relative accessibility scores in female donor over 5 runs.
- **Table S4:** Precision, recall, F1 achieved by the various de novo motif discovery approaches in retrieving TFs from ATAC-seq peaks of the two cell types. Expressed TFs (from RNA-seq data) that intersect with our HT-SELEX set of TFs are used as the gold-standard for retrieval. **(a)** Performance with default $p$-value and e-value cut-offs for all methods. **(b)** For the two best approaches, HOMER and BindVAE, performance upon varying the cut-offs is shown. The performance of a naive match-every-motif classifier (label everything positive) is shown.
- **Figure S9:** Select latent dimensions (topics) from the GM12878 model that are referred to in the main text. The shown matrix is a sub-matrix of $\theta \in \mathbb{R}^{M \times D}$. Dimensions 69 and 83 are redundant, in that they both assign high weights to the same k-mer features.
- **Figure S10:** Redundant dimensions: Latent dimensions with similar k-mer distributions are shown. In each plot, the two dimensions were mapped to the same TF; for example: #69 and #83 are both mapped to HEY1. Along the x-axis is the union of the top 1000 8-mers from both dimensions. The values in each row are the decoder parameters learned by the model: $\vec{\theta}_i$ for the $i^{th}$ dimension.
- **Figure S11:** GM12878 peaks projected onto the 10 noisy latent dimensions.
- **Figure S12:** Interpreting the models learned on random regions of the DNA. **(Top)** Matrix showing the learned decoder parameter $\theta$ (that shows the k-mer distributions for each latent factor) for a model trained on 150,000 random regions of naked DNA from mouse E16.5 sorted germ cells. **(Bottom)** Matrix showing the learned decoder parameter $\theta$ for a model trained on flanking genomic regions from GM12878 cells, where regions that are 10kb away from the ATAC-seq peaks were chosen as training data.
- **Table S5:** TFs learned by the model trained on the GM12878 ATAC-seq data.
- **Section 2:** Description of latent factors capturing non-TF related patterns.
- **Section 3:** Details of the comparison to other baselines.
- **Section 4:** Further details on datasets.
- **Figure S13:** GM12878 peaks that contain sequences with the most repeats (top 5 are shown), as ranked by the Tandem Repeat Finder score (TRF algorithm [41]). We find that the highest-scoring topic for all of these sequences is the same. The genomic coordinates correspond to the `hg19` assembly.
- **Figure S14:** A549: Heatmap showing the top 20 k-mers learned by our model for each latent dimension.
- **Figure S15:** A549: Heatmap of the latent space obtained by our A549-trained model, upon doing inference on 11,600 SELEX probes from 51 TF experiments. Each row is the latent representation $\vec{z}$ of a HT-SELEX probe, with the rows being colored by the TF experiment that the probe comes from. There are 200 enriched probes per TF/HT-SELEX experiment.
- **Figure S16:** PCA of the k-mer distributions in two isogenic replicates of T cells.
- **Figure S17:** Box-plots showing the distribution of topic scores for ATAC-seq peaks that overlap with ChIP-seq peaks and those that do not, for several TFs.
- **Figure S18:** Box-plots showing the distribution of topic scores for ATAC-seq peaks that overlap with ChIP-seq peaks and those that do not, for several TFs.

**Additional file 2.** Review history.

## Acknowledgements
We would like to thank William Noble for discussions and Michael Figurnov for help with the implementation of the VAE.

## Peer review information

## Review history
The review history is available as Additional file 2.

## Authors' contributions
MK worked on problem formulation, method development and implementation, data processing, generating plots, and writing. HY contributed to problem and analysis discussion, data preprocessing, generating plots, and writing. JLF contributed to problem formulation, method discussion, and editing. CL contributed to problem formulation, analysis discussion, designing experiments, and writing. All authors read and approved the final manuscript.

## Authors' Twitter handles
- Meghana Kshirsagar: https://twitter.com/meghanaksagar
- Han Yuan https://twitter.com/HY3952
- Juan Lavista Ferres: https://twitter.com/BDataScientist

## Funding
This research was supported by Microsoft, Calico, and NIH/NHGRI award U01 HG009395 to CL.

**Availability of data and materials**
**Implementation**
The source code, sample data and data processing scripts are available at: https://github.com/microsoft/BindVAE/ [42].

**Data and model downloads**: The source code, trained models, ATAC-seq datasets, HT-SELEX probes after QC and processed features are available for download on Zenodo at `DOI:10.5281/zenodo.6658242` here: https://zenodo.org/record/6658242 [43].

**Datasets**
We downloaded and used the following publicly available datasets, which are explained in detail in the Methods section.

**ATAC-seq data:**

- GM12878: GEO accession number GSE47753 https://www.ncbi.nlm.nih.gov/geo/query/acc.cgi?acc=GSE47753
- A549: https://www.encodeproject.org/files/ENCFF548PSN/
- T cell female adult: https://www.encodeproject.org/experiments/ENCSR977LVI/
- naive B cell female donor: https://www.encodeproject.org/experiments/ENCSR685OFR/
- naive B cell male donor: https://www.encodeproject.org/experiments/ENCSR903WVU/
- naive CD8+ T cells mouse: https://pubmed.ncbi.nlm.nih.gov/33891860/
- naive CD8+ T cells human: https://pubmed.ncbi.nlm.nih.gov/33891860/

**HT-SELEX data from Jolma et al. [18] and Yang et al. [39]:** ENA accession numbers ERP001824 and ERP001826 available at: https://www.ebi.ac.uk/ena/browser/home.

**CAP-SELEX data from Jolma et al. [20]:** Available at the ENA: http://www.ebi.ac.uk/ena/data/view/PRJEB7934

**ChIP-seq data: GM12878**: The data downloaded from ENCODE https://www.encodeproject.org/ is also available in the `datasets/` folder of the github repository.

**RNA-seq datasets:**

- GM12878: https://www.encodeproject.org/files/ENCFF906LSJ/, https://www.encodeproject.org/files/ENCFF630BDD/
- A549: https://www.encodeproject.org/files/ENCFF203NNS/
- T cell female donor: https://www.encodeproject.org/experiments/ENCSR336VTK/

# Declarations

**Ethics approval and consent to participate**
Not applicable.

**Consent for publication**
Not applicable.

**Competing interests**
MK and JLF are employees at Microsoft. HY is an employee at Calico Life Sciences.

**Author details**

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

## 