## [**Additional file 2** Review history. · Genome Biology]

Review History

First round of review

Reviewer 1

Were you able to assess all statistics in the manuscript, including the appropriateness of statistical tests used? Yes: Please describe the values under the motifs in Figure 3 A and B, are they p-values, if so, what are they comparing?

Were you able to directly test the methods? No.

Comments to author:

Please carefully check your figure legends. For example, Figure 1 is missing part d.

Only one GM12878 ATAC-seq dataset was used to represent DNA accessibility, how reliable is this?

For the ChIP-seq data, please specify how many ENCODE ChIP-seq data was used and whether you excluded any samples that were under a treatment.

Are there differences in motifs between different cell lines and tissue types?

Do you have any thoughts on how DNA variants (SNPs or mutations) could influence TF binding across other cell lines (perhaps in patients)?

In the discussion, please elaborate on the applicability and significance of identifying expressed TFs / TF occupancy. I understand that it can help decode gene regulatory programs, but how?

Reviewer 2

Were you able to assess all statistics in the manuscript, including the appropriateness of statistical tests used? Yes: In the results section "BindVAE: a Dirichlet[...]", the authors should explain and justify the use of wildcards in the k-mer representation. Also in this section, the authors briefly touch on the role of low-complexity regions and cleavage bias, but other than the statistical test in algorithm 1, they do not address how these components can be separated from TF binding signal. This seems to be a key advantage of Dirichlet autoencoders, and so the authors should emphasize that the topics learned are not distributed mixtures of different signals, such as a TF that has a binding motif that overlaps with cleavage bias.

Were you able to directly test the methods? No.

Comments to author:

Summary

Here the authors seek to show how unsupervised deep learning methods can be leveraged to interpret the relationship between sequence and chromatin accessibility. The authors use a Dirichlet variational autoencoder (VAE) called BindVAE to create a regularized latent space representation of the role of k-mers enriched across GM12878 ATAC-seq peaks. The regularization component of BindVAE allows for this latent space to have the capability to not only represent motifs, but represent combinatorial relationships between motifs. The authors show that BindVAE produces latent dimensions which contain k-mers mapping to known transcription factor (TF) motifs and are enriched for in vitro TF binding (measured by HT-

SELEX). Motifs can be reconstructed from the trained 8-mers into more complex motifs, some of which are found to represent cooperative binding between two TFs (e.g. FOXJ3 and TBX21). The authors show that when comparing the ATAC-seq latent dimensions to in vivo ChIP-seq data, the scored latent dimensions can distinguish ChIP-seq peaks of different TFs. Using these results, the authors show that BindVAE can learn motifs from k-mer enrichments, using these to distinguish different TF binding events.

General comments

This study provides an excellent proof-of-principle of how unsupervised deep learning may be applied towards understanding how genomic sequence drives chromatin accessibility. By allowing the latent dimensions to characterize descriptive k-mer combinations and then applying HT-SELEX to incorporate TF binding, the authors have created a way of uncovering an alternative representation of sequence grammar that is more combinatorially informative than a set of PWMs. However, while the manuscript provides elegant and novel methodology to discover meaningful sequence representations in an unsupervised manner, there is insufficient discussion about how BindVAE provides novel or practical advantages towards the application of de novo motif discovery using ATAC-seq. Additionally, there is insufficient benchmarking and use of controls to convince the reader that this tool's novelty allows for greater understanding or confidence towards characterizing motif grammar than previously published methods. These latent dimensions should be able to characterize combinatorial k-mer enrichments in interpretable latent space, which seems to be a key strength of this method, but the manuscript does not sufficiently explore this capability. Overall, this is an exciting application of unsupervised learning towards genomics data, but the scope of the manuscript must extend past "proof-of-principle methodology" in order to provide strong value towards the field.

Specific points for the authors

Overall, the capabilities of these latent dimensions to (1) identify motifs, (2) correlate well with HT-SELEX in vitro binding, (3) represent motifs that are indicative of co-binding, and (4) distinguish between different TF ChIP-seq peaks is impressive, elegant, and exciting to think about. However, a general weakness of this manuscript is that while the technical details are described very well, there is little biological discussion throughout the manuscript body when discussing each of these four key features of BindVAE. This makes it very hard for a biological reader to read the results as meaningful and biologically applicable. "If we find X mathematical representations using our method, this could be interpreted biologically as Y." For papers that apply highly-complex computational methods towards biological applications, this descriptive language is key towards conveying the applicability and novelty of your interpretations.

There seem to be three key limitations to BindVAE that are not discussed in the manuscript. (1) "Bag-of-words" inputs remove positional information from the k-mers across a genomic window. (2) "Bag-of-words" inputs remove magnitude information from the ATAC-seq peak (whether it is very accessible or less accessible). (3) Mammalian genomes contain repetitive elements (retrotransposons/ERVs) that are functionally bound by TFs (e.g. Wang et al, 2020, Cell Regeneration, <https://doi.org/10.1186/s13619-020-00046-4>) in order to regulate biological processes. These highly repetitive elements would cause commonly recurring k-mer enrichments

that might overrepresent latent dimensions. The manuscript would be strengthened by comprehensive discussion about the limitations of interpretability of the results as well as the correction steps for possible genomic artifacts.

The introduction is appropriate for the scope of the manuscript, however would be strengthened by additional citations. Paragraph 2 should be supported by citations. Paragraph 6 (last paragraph) would be strengthened by adding more precise explanations of why unsupervised deep learning provides advantages that supervised deep learning does not, in addition to mentioning that unsupervised learning is "under-explored".

In the results section "BindVAE: a Dirichlet[...]", the authors should explain and justify the use of wildcards in the k-mer representation. Also in this section, the authors briefly touch on the role of low-complexity regions and cleavage bias, but other than the statistical test in algorithm 1, they do not address how these components can be separated from TF binding signal. This seems to be a key advantage of Dirichlet autoencoders, and so the authors should emphasize that the topics learned are not distributed mixtures of different signals, such as a TF that has a binding motif that overlaps with cleavage bias.

In the "BindVAE learns diverse [...]" section, the authors point out that the theta matrix is diagonal-heavy, and use this to suggest that the learned dimensions represent different TFs. It would be informative to see this same matrix generated on genomic background regions, i.e., regions lacking ATAC-seq peaks. The authors should also provide an illustration of dimensions "that capture highly redundant information", and provide some discussion of what those dimensions capture.

In "Motifs discovered de novo[...]", there is reference to "obtain[ing] the top-200 10-mer sequences that scored the highest", but it is not clear how they are calculating the score. Is this simply the value of the latent dimension when the input contains only the 8-mers generated from the candidate 10-mer? Is it some sort of attribution to individual 10-mers? The authors claim that their model "is biased toward learning shorter motifs more accurately.". The authors must demonstrate that their motifs are more accurate than the motifs in CIS-BP.

Throughout the manuscript there are interpretations surrounding latent dimension "scores" that refer to Algorithm 1. Please clarify in the methods what is considered a "score" and use consistent wording throughout the Methods and the manuscript body so the reader has a clear idea of what is used as latent dimension scoring. Additionally, please explicitly mention whether the TF probes are from the HT-SELEX experiments in the Methods description of Algorithm 1.

In the manuscript body under "Mapping the TFs to dimensions", paragraph 2 claims that some latent dimensions are not represented by TFs and conclude that they represent genomic background. Here, the manuscript would be strengthened by further explanation of this statement. How many latent dimensions of the 100 total represent genomic background? How many of them represent Tn5 sequence bias? If they represent genomic background, are there low-complexity k-mers (CG-rich/AT-rich) that keep recurring?

In "mapping peaks to TFs", it is not clear why the authors assign each peak to a single TF. As

they illustrate, multiple TFs bind to every ATAC peak, and the paper would be strengthened by representing each peak by its ensemble of TF signals instead of its maximal signal.

In the manuscript body under "Projecting HT-SELEX probes into the latent space", the language in paragraph 1 describing the rationale for using HT-SELEX needs to be expanded upon. Earlier in the manuscript, it has been strongly shown that the VAE is able to learn consensus patterns of TF binding motifs (linking VAE latent dimensions \leftrightarrow discovered motifs). The addition of HT-SELEX is meant to bring in vitro binding TF information into play to validate that these motifs are indeed meaningful to TF binding (linking VAE latent dimensions \leftrightarrow in vitro binding). This needs to be described more clearly, with the rationale being laid out prior to describing the heatmap of Figure 2B. Furthermore, the biological meaning of this finding should be explored. For example, do these similar TFs tend to bind in the same places (implying that the model has correctly mapped them into the same latent dimension since they represent similar biological processes), or do these similar TFs have distinct binding profiles, suggesting that the model has oversimplified its latent space? It would also be informative to compare the learned motifs and PWMs for one of these families of TFs.

In "top 8-mers learned [...]", the authors should address the role of reverse-complementation in their sequences. They compare GGGGA to TTCCC and claim that these represent different ends of the motif, but they are nearly reverse complements of each other (since the motif is essentially palindromic). The authors should carefully choose a motif that is not palindromic if they wish to claim that a single latent dimension has captured multiple parts of a motif.

In "Cooperative binding signals", the authors show that one particular dimension seems to capture two TFs. By comparing the activation of that dimension given sequences that bind to the two TFs individually, they show that sequences designed to bind to both simultaneously are more active in that dimension. This claim would be strengthened with appropriate controls on the k-mer distributions of the input sequences. For example, the CAP-SELEX protocol could be more efficient than HT-SELEX, leading to a generally more motif-rich final sequence. The authors should ensure that the degree of enrichment between the two experiments is the same, and they should choose a pair of TFs that have distinct binding motifs. If the two TFs have very similar binding motifs, it is unsurprising that CAP-SELEX would introduce two similar motifs into its designed sequence. If the corresponding latent dimension cannot distinguish between the motifs, then it will naturally give a high value when its motif is enriched in the input.

In "Accessibility patterns predicted[...]", the authors must demonstrate that their model gives similar results for similar cell types, rather than running two drastically different cell types. It would be reassuring to see two datasets collected for the same cell type by different experimental groups to know that the different activation patterns are due to actual biological differences.

In "Comparison with HOMER", the authors should assess the quality of the motifs identified by HOMER. As seen in Figure 1, the motifs from BindVAE are often different than the HT-SELEX PWMs. Are the HOMER motifs more similar to the HT-SELEX ones? It would also be good to compare motifs derived from other, more recent, approaches.

The methodology of how the "accessibility score" is computed in the manuscript body (linked to

Figure 3D) should be moved to the Methods section. Additionally, this segment on accessibility patterns would be strengthened by deeper discussion into the findings. This is an opportunity to explore differences between cell types in order to biologically validate your computational methods of de novo motif discovery. Do these results make sense? For TFs with similar accessibility scores, do they contain similar consensus k-mer reconstructions or are there differences?

In the discussion, the authors claim that CNN-based methods do not generalize across cell types. It is not clear that the method in this paper is able to generalize across cell types short of being re-trained on each different cell type. For example, if there is a TF that is found in cell type B, but not cell type A, and the model is trained on type A, there is no expectation that the B-specific TF will be found. CNN-based models must, by their nature, be trained on every cell type of interest, and this is a limitation of these methods. If the authors have developed a method that can be trained once and then allow for interpretation of data from different cell types, they should clearly illustrate this useful finding.

The figures in this manuscript need to be revised, as many are missing axes labels, have mismatching figures/figure descriptions, and do not clearly convey the points the manuscript body is describing. Below are a list of suggestions:

Table 1 description contains a misspelling: "chip-seq" should be "ChIP-seq".

Figure 1B's description is confusing. Please rephrase to convey the meaning of the order of k-mers that you are representing ("sorted by their value", is it sorted by the latent dimension weights or the value of the kmer?). There is a misspelling "entires", which I presume you mean "entries". Additionally, you refer to latent dimension 0 and 72 in the body of the manuscript, but it is very difficult to see in the figure itself. Can you annotate these dimensions to increase clarity of this figure? Though the plot illustrates the top 20 k-mers for each dimension, it seems that more than 20 pixels are non-zero for each latent dimension. The authors should provide a color bar with a distinguished zero value.

Figure 1C would be easier to read if you match the "Learned motif" orientation to the "CIS-BP" orientation. CIS-BP makes both orientations available in their database, so this should be feasible. It is also recommended that you create two rows for TFs that contain 2 latent dimensions (e.g. CTCF and RUNX3) instead of condensing the logos. This will allow viewers to more clearly see the PWM representations to compare them to the CIS-BP database entries.

Figure 1 should also include some information to support the claim that "We analyze some of these dimensions and find that they represent genomic background".

Figure 2A and 2B are missing axis labels, please add to improve clarity of figure. Additionally, Figure 2B does not contain clear labels of what the colors on the heatmap represent. Please label the plot and describe in clearer detail.

Figure 3C would be easier to understand by reiterating in the graphic itself that you are showing a FOX3J-TBX21 cooperative motif by labeling it as such. Additionally, Figure 3A and 3B would be clearer to understand if you projected the logos of each motif above the barplots to give the readers a clear understanding of the motifs you are trying to show cooperative relationships between.

Figure 3D is missing y-axis labels, the key occludes the data and should be shrunk. The description of the "accessibility score" in the manuscript body does not match the labeling on the

x-axis. Please clarify so that this matches.

Figure 3E is missing the Expressed-only value.

Figure 4A and B would be strengthened by also showing the different TF binding motif consensus for each TF classification. This suggestion applies particularly to SPI1 and IRF4 in Figure 4B due to the poorer clusters formed by UMAP. Are these two motifs highly similar? Why is IRF4 having trouble being distinguished? Are these motifs highly cooperative? What is the overlap of ChIP-seq peaks themselves (i.e. are IRF4 and SPI1 ChIP-seq peaks overlapping in vivo as well? Are the latent dimension ambiguities shown by poor UMAP consensus clusters reflecting something we see in vivo as well?)

Figure 4C is missing axis labels. Also currently the authors are claiming "The latent dimensional scores based on ATAC-seq enriched k-mers can distinguish different TF binding peaks."

However, this is not strongly conveyed, because the reader does not know which TF ChIP-seq peak is connected to which row. Presumably, we can assume that the top 66% of rows are attributed to RUNX3, but we can't be sure because that information is not available. If you can add an extra column/annotation that can connect the row with its TF identity, this figure would be even stronger. This is particularly true because it appears that NRF1 seems to contain co-binding relationships with RUNX3 and MAX in some cases, but since we cannot be sure of each TF's identity, then this is difficult for the reader to assume.

In Algorithm 1, there is a reference to the variable z_{-t}^s , and this is not defined clearly.

Reviewer 3

Were you able to assess all statistics in the manuscript, including the appropriateness of statistical tests used? Yes: see review.

Were you able to directly test the methods? No.

Comments to author:

Overall assessment

A comprehensive characterization of mammalian TF binding motifs remains unsolved. In the last years, deep learning based approaches have yielded impressive progress on sequence-based modeling of omics assays. However, their mechanistic interpretation in terms of individual TF binding remains tedious. Here the authors contribute to this research area with an interesting approach that leverages the deep learning machinery, but which, by design, allows a more straightforward interpretation in terms of TF binding motifs.

Specifically, the authors describe BindVAE, a novel unsupervised deep learning method to jointly learn transcription (TF) binding patterns from open chromatin regions. BindVAE is a Dirichlet variational autoencoder which encodes a kmer representation into a latent space where many latent dimensions, according to the authors, map to specific TFs. It is shown that the highest scoring kmers for some latent dimensions can be assembled into known motifs and claimed that other latent dimensions capture the genomic background present in open chromatin regions. Furthermore, the authors employ an algorithm that allows assigning latent dimensions to TFs by scoring HT-SELEX probes with known binding sites which they later verify by projecting ChIP-seq peaks into the latent space. For some latent dimensions they find kmer patterns from multiple TFs which could suggest cooperative binding. They try to verify their assumption by showing that CAP-SELEX probes from cooperative binding motifs show higher

activation of a latent dimension than the separate HT-SELEX probes.

Overall, there are major revisions needed as detailed below. Regarding the impact, the manuscript at this point is interesting for the computational modeling community. For the practitioners, it is hard to tell whether this tool can be substituted to existing tools, due to lack of thorough and extensive benchmarks.

Major points

1. A detailed and mathematically clear description of the model is important. To avoid ambiguous text sections in the Methods section, we suggest introducing clear notations, formulas for the definitions like model loss, input vector of the model, and output vector.
2. The authors claim that some latent dimensions recover cooperative binding. "Since the co-occurrence of multiple binding patterns might merely suggest that binding sites of two different TFs are present in a peak and not necessarily cooperative binding" the authors use sequences from CAP-SELEX experiments and show that the activation the latent dimension of interest is higher for CAP-SELEX sequences than for the HT-SELEX sequences from the two separate TFs. We have the following concerns. First, the procedure of how the two shown examples are chosen is not clear. What made dimension 60 and 67 distinct? This feels cherry-picked. Any overall statistics?
3. Second, the authors need to prove that the higher score is not merely because the CAP-SELEX sequences has both motifs in it. So the adequate comparison would be CAP-SELEX sequences compared to sequences which contain the motifs of both TFs.
4. Third, the authors compare the activation of FOXJ2 and FOXL1 to FOXJ3-TBX21 and not FOXJ3.
5. Fourth, the authors show an example of how the cooperative binding motif looks for FOXJ3-TBX21 but not for MYBL1-MAX.
6. The authors claim that they can predict accessibility patterns for GM12878 and A549 by using the summed activation over all peaks for each latent dimension. While it is not surprising that the latent activations differ between cell types (as they have been trained on separate sets of ATAC-peaks) there is also no comparison to what factors are known to be active in those cell types.
7. The authors compare their identified TFs to algorithm HOMER. One issue is that HOMER fits PWMs and then does a matching against a provided set of PWMs. It is unclear whether the difference between HOMER and BindVAE comes from the PWM fitting algorithm, which corresponds to the fitting of the autoencoder, or from the PWM matching algorithm, which corresponds to BindVAE's post-processing Algorithm 1. The authors should provide an analysis that delineates both contributions.
8. A second issue with the comparison with HOMER is that it is done for fixed cutoffs. The authors should provide instead "continuous" comparisons, e.g. using ROC or PRC curves.
9. A third issue is that the comparison is limited to only one motif finding algorithm. A review of existing motif enrichment algorithms should be done and the most prominent ones should be compared to. I am expecting MEME, GLAM2, and probably GADAM. Eventually, the performance should be put in perspective of overall compute time for the different methods.
10. A fourth issue is that the Methods section provides no detail on how HOMER was applied.
11. With Figure 4's analysis, the authors aim to show that the latent dimensions correspond to known TFs by using ChIP-seq peaks for the identified TFs. The authors find that only 12 TFs overlap between HT-SELEX and reliable ChIP-seq data but do not specify what a reliable ChIP-seq data means in that context (all of them come from ENCODE).
12. Furthermore, the representation chosen (Figure 4c) to prove the claim is not nailing the point. One could think of plotting a barplot where they compare latent activations of peaks with ChIP-seq signal to all other peaks. Alternatively, one could also phrase it as a prediction task by evaluating how well a latent dimension can distinguish peaks with that TF bound (overlap with ChIP-seq) from all other peaks. Evaluation metric would be AUPRC.

Minor points

-
13. The multinomial distribution is defined for counting random n independent trials over k exclusive categories. However, the trials are not independent (due to overlap of the substrings which tile the region). Moreover, the categories are not exclusive due to the wildcard kmers. The model may "do the job" anyways - it's just a loss after all - but this point should be discussed.
 14. How many TFs are mapped to each latent dimension with Algorithm 1? It would be helpful to have an overview as a table for example.
 15. It is not clear how the claim is made that dimensions that can not be mapped to TFs are genomic background. How was this analysis done?
 16. Figure 2b, TF names are not aligned. It would improve the legibility of the figure a lot if the legend was on the left next to the actual color. This would make it a lot easier to know which TF corresponds to which row. This is especially true for people who are colorblind and can not distinguish the colors.
 17. Page 4: "and assume we" is missing words.
 18. Figure 3d: The gene names are capped.
 19. Table 1: The legend should say ChIP-seq instead of chip-seq.
 20. Part: "Projecting ChIP-seq peaks into the latent space". The authors continuously talk about 12 TFs but in Figure 4c they only show 11.
 21. Figure 4a/b: It would be nice to choose a colorblind-friendly palette as green and orange are very difficult to distinguish.
 22. Algorithm 1. Is the shift (typically denoted " μ ") for the Null hypothesis in the Mann-Whitney U test different from 0 ? How is it computed? Please provide details.

Dear Editors,

We would like to thank all the reviewers for their time and efforts in giving us detailed and insightful comments that have helped improve the manuscript by leaps and bounds. We have tried our best to address all the concerns, in particular regarding benchmarking and biological relevance. Below, we answer the major questions point-wise and provide data and plots to support the response.

Reviewer #1:

- 1 ***Only one GM12878 ATAC-seq dataset was used to represent DNA accessibility, how reliable is this?***

We now show detailed results (qualitative: several figures and quantitative: **Supplemental Table S3(a), S3(b)**) on three ATAC-seq datasets: GM12878, a B lymphoblastoid cell line, and A549, a lung epithelial cell line and a T-cell sample from an adult female. Additionally, we show qualitative analyses of three other datasets: mouse and human CD8 T-cells, naïve B cells from two donors, T-cell samples from two donors (**Supplemental Figure S8**).

- 2 ***For the ChIP-seq data, please specify how many ENCODE ChIP-seq data was used and whether you excluded any samples that were under a treatment.***

We download all available conservative and optimal IDR (irreproducible discovery rate) thresholded 'narrowpeak' files of ENCODE GM12878 ChIP-seq data from the ENCODE portal. We do not exclude any samples based on their treatment. The analysis is done only on TFs that overlap with the TFs represented in the HT-SELEX data set, as described in the paper.

- 3 ***Are there differences in motifs between different cell lines and tissue types?***

As shown in the paper, we find cell-type specific differences in the TFs whose binding signals are recovered by BindVAE, consistent with the different TFs expressed in these cell types (see **Figure 3d** in main manuscript). There is some evidence that we also find differences in motifs for the same TFs between cell lines, in cases where these TFs are expressed in both. We show some examples in the table below for the two cell lines GM12878 and A549.

We also compare all the k-mer distributions learned for both cell types by visualizing them together in the PCA plot shown below. We apply PCA on the decoder parameters $\theta \in \mathbb{R}^{100 \times 112800}$ that capture the weights of each of the 112800 k-mers in each of the latent dimensions $i = [1, \dots, 100]$, for both cell types together.

TF	GM12878 motif	A549 motif	CIS-BP motif
HNF4A			NFIA			SRY			ELF5			OLIG3			
Table R1 [also included as **Supplemental Table S1**]. Motifs constructed from latent factors learned for GM12878 and A549 for TFs learned by our model. HNF4A, NFIA, SRY have the same `accessibility score' in both GM12878 and A549 trained models (see Figure 3d in the main paper). ELF5 and OLIG3 are both expressed in both cell types.

Figure R1 [also included as **Supplemental Fig. S8**]. PCA of the 100 BindVAE k-mer distributions (given by the decoder parameters $\theta \in \mathbb{R}^{100 \times 112800}$), from models trained on GM12878 and A549. The

PCA shows a cluster of topics at the bottom, possibly capturing similar patterns and some distinct patterns being captured by the topics in the semicircular regions at the top.

- 4 ***Do you have any thoughts on how DNA variants (SNPs or mutations) could influence TF binding across other cell lines (perhaps in patients)?***

This is a great question, and we have considered this as a future application of the model. In order for the model to be sensitive to single base-pair level changes (i.e. SNPs), we would need a different input representation that does not rely on k-mer wildcards that are designed to allow for mismatches. Therefore, the extensions needed to represent genetic variation in the model are outside the scope of the current paper.

- 5 ***In the discussion, please elaborate on the applicability and significance of identifying expressed TFs / TF occupancy. I understand that it can help decode gene regulatory programs, but how?***

Broadly speaking, the regulatory genomics field uses chromatin accessibility as measured by ATAC-seq to map candidate gene regulatory elements, including enhancer elements for genes. We note that not all accessible elements are enhancers, and not all elements involved in regulation of gene regulation are accessible (e.g. repressive elements may not be associated with open chromatin). Acknowledging these caveats, the next steps in decoding gene regulatory programs are to (i) associate ATAC-seq peaks with target genes and (ii) identify the TFs that might be binding peak regions and therefore regulating target genes. Problem (i) is not addressed here but is the subject of a wide range of current research activities, including the use of chromosome conformation capture assays to map 3D promoter-enhancer interactions and of single-cell multiomic data to enable correlation of peak accessibility and gene expression across individual cells. Problem (ii) is what we address here, by decoding the TF binding signals in specific candidate regulatory elements (peaks). Solving both problems will lead to mechanistic insight into the TF networks that regulate individual genes and gene expression programs. We have clarified this point in the Discussion.

Reviewer #2:

- 1 ***While the manuscript provides elegant and novel methodology to discover meaningful sequence representations in an unsupervised manner, there is insufficient discussion about how BindVAE provides novel or practical advantages towards the application of de novo motif discovery using ATAC-seq. Additionally, there is insufficient benchmarking and use of controls to convince the reader that this tool's novelty allows for greater understanding or confidence towards characterizing motif grammar than previously published methods. These latent dimensions should be able to characterize combinatorial k-mer enrichments in interpretable latent space, which seems to be a***

key strength of this method, but the manuscript does not sufficiently explore this capability. Overall, this is an exciting application of unsupervised learning towards genomics data, but the scope of the manuscript must extend past "proof-of-principle methodology" in order to provide strong value towards the field.

We thank the reviewer for finding the method to be “elegant and novel” and an “exciting application of unsupervised learning towards genomics data”. We have added substantial additional analyses and results to try to address critiques related to demonstrating the practical advantages the approach, expanding the benchmarking against other methods, and in general extending beyond a “proof-of-principle” methodology. As appreciated by the reviewer, BindVAE uses a k-mer distribution-based approach that allows the model to capture more information than PWMs, which assume that successive positions within the motif are independent of one another. We have now benchmarked the performance on retrieving binding sites from expressed TFs by comparing with 3 other approaches: HOMER, MEME, GADEM (described in response to Reviewer #3, question 9). In addition to our main results on ATAC-seq from two ENCODE cell lines, GM12878 and K562, we have run BindVAE on human B cell and T cell data from pairs of donors to demonstrate stability of the model. We also explore in more detail the extent to which latent dimensions can disentangle low complexity or repeat k-mer signals from TF binding signals, and whether transposase sequence preferences are captured as latent dimensions. We hope that these additional results now demonstrate BindVAE’s strong value to the field.

- 2 ***There seem to be three key limitations to BindVAE that are not discussed in the manuscript. (1) "Bag-of-words" inputs remove positional information from the k-mers across a genomic window. (2) "Bag-of-words" inputs remove magnitude information from the ATAC-seq peak (whether it is very accessible or less accessible). (3) Mammalian genomes contain repetitive elements (retrotransposons/ERVs) that are functionally bound by TFs (e.g. Wang et al, 2020, Cell Regeneration, <https://doi.org/10.1186/s13619-020-00046-4>) in order to regulate biological processes. These highly repetitive elements would cause commonly recurring k-mer enrichments that might overrepresent latent dimensions. The manuscript would be strengthened by comprehensive discussion about the limitations of interpretability of the results as well as the correction steps for possible genomic artifacts.***

(1) Firstly, while we lose the exact positional information due to our choice of bag-of-words as input, we do not lose any context information as the surrounding DNA sequences are still part of the input representation; for this broader sequence context, the exact position of a specific k-mer in the 200bp peak may not be important for a successful encoding. Secondly, the bag-of-words representation

actually gives us flexibility – the input DNA sequence can be of arbitrary size, and we can apply the same model on inputs of varying size from diverse *in vivo* and *in vitro* data. This is not true for a one-hot-encoding based input representation used by other deep learning models, which requires the DNA sequence inputs to be of a fixed constant size. We find that there is a trade-off between the complexity of the input representation and the interpretability of the resulting model. Our choice of a bag-of-words input makes the model readily interpretable by giving us an easy way to map components of the latent space to distributions over k-mers. Lastly, it would be straightforward to add positional information in our model the way transformers do, by adding position encodings. We now include a description of this extension in the Discussion.

- (2) Our model does not use the real-valued peak accessibility as an input and therefore does not capture the relative frequency of binding patterns that are in less accessible regions vs. more accessible regions. However, we note that many computational methods for finding TF binding signals in bulk ATAC-seq make the same simplifying assumption of treating reproducible peaks as positive examples – without retaining accessibility values – and perhaps using inaccessible regions or shuffled genomic sequences as a negative background. For example, virtually all PWM-based motif finders (HOMER, MEME) are blind to real-valued accessibility information, and even the widely-used discriminative model gkm-SVM trains with binary rather than real-valued labels. That said, one straightforward extension to include accessibility information in BindVAE’s k-mer count based input would be to scale the counts based on accessibility of the region that each k-mer comes from. We now mention this potential extension in the Discussion.
- (3) We agree that repeat elements and low complexity regions that overlap reproducible ATAC-seq peaks may confound both unsupervised and supervised sequence models, since they may manifest as overrepresented k-mers/motifs that do not represent TF binding signals. To explore this issue, we examined the latent representation for the top 30 low-complexity peak regions as identified by the Tandem Repeat Finder algorithm and observed that a low-complexity signature was captured by a specific latent dimension in the model. Please see **Figure R2-R3** below for details. This suggests that BindVAE may indeed be able to disentangle TF binding site signals from repeat-associated enriched k-mer patterns. These results are now included as **Supplemental Fig. S5 and S3** and discussed in the manuscript.

- 3 ***The introduction is appropriate for the scope of the manuscript, however would be strengthened by additional citations. Paragraph 2 should be supported by citations. Paragraph 6 (last paragraph) would be strengthened by adding more precise explanations of why unsupervised deep learning provides advantages that supervised***

deep learning does not, in addition to mentioning that unsupervised learning is "under-explored".

We have added citations for these paragraphs in the Introduction. Our paper explores unsupervised deep learning approaches for learning TF binding signatures from DNA sequences using chromatin accessibility data alone, without requiring large-scale TF-specific experimental data. Naturally, it is possible to train discriminative deep learning models on sequence data to predict chromatin accessibility either as a binary label (open/closed) or in the regression setting, as many authors have done (e.g. the Basset and Basenji models from David Kelley's group). However, interpreting supervised models requires attribution of the learned parameters to the output labels – which is often not a robust process as many works in the broader deep learning literature have shown. For example, in medical image analysis, image features that are irrelevant for clinical analysis of the specimen can be used by the model to improve the prediction accuracy based objective function. In regulatory genomics, methods to interpret the sequence information captured by deep discriminative sequence models (e.g. DeepLIFT) or to improve the interpretability of these models (e.g. use of a Fourier prior in training) are still in their infancy.

Generative models, such as VAEs, are naturally more interpretable by design. For example, BindVAE seeks to capture individual TF binding signals as latent dimensions in the model, as well as disentangled other sequence signals like low-complexity k-mer signatures. More broadly, generative models learn universal representations of the data that can then be used in downstream task-specific applications. We propose that such methods are the next step in the evolution of machine learning modeling on genomics data since supervised models are typically not transferable to other prediction tasks. We have summarized these advantages of generative modeling in the Discussion.

- 4 ***In the results section "BindVAE: a Dirichlet[...]", the authors should explain and justify the use of wildcards in the k-mer representation. Also in this section, the authors briefly touch on the role of low-complexity regions and cleavage bias, but other than the statistical test in algorithm 1, they do not address how these components can be separated from TF binding signal. This seems to be a key advantage of Dirichlet autoencoders, and so the authors should emphasize that the topics learned are not distributed mixtures of different signals, such as a TF that has a binding motif that overlaps with cleavage bias.***

In the manuscript body under "Mapping the TFs to dimensions", paragraph 2 claims that some latent dimensions are not represented by TFs and conclude that they represent genomic background. Here, the manuscript would be strengthened by further explanation of this statement. How many latent dimensions of the 100 total

represent genomic background? How many of them represent Tn5 sequence bias? If they represent genomic background, are there low-complexity k-mers (CG-rich/AT-rich) that keep recurring?

Wild-cards in k-mers: This modeling choice is motivated by extensive previous work in our group, including BindSpace (Yuan et al., Nature Methods 2019), SeqGL (Setty et al., PLOS Computational Biology 2015), and our early string kernel work for epigenomic data (e.g. Arvey et al., Genome Research 2012) that anticipated the widely-used gkm-SVM method (Ghandi et al., PLOS Computational Biology 2014). All these approaches use short k-mers (e.g. k=8) with some kind of inexact matching, such as the use of wildcards. Because TF binding signals are degenerate, the use of wildcards that capture a larger number of binding instances has consistently proven useful in machine learning models of regulatory sequences.

Low-complexity regions and genomic background: We used Tandem Repeat Finder (TRF) to find GM12878 peaks that contain low-complexity regions (LCRs) or repeats. We found that the top scoring ~30 peaks (scores assigned by TRF) contain repeats throughout the 200bp region, while the subsequent peaks contain some TF binding sites in addition to some repeat regions. We found that the latent representation learned for these top LCRs or regions with repeats are similar as shown by the heatmap below (**Figure R2**), where most of them have a high value for the same latent factor (#37). We show the top 100 k-mers from latent factor #37 in **Figure R3** below and they do contain recurring CG-rich k-mers as guessed by the reviewer.

Figure R2. [also **Supplemental Fig. S5**] Latent representations z for the top low complexity GM12878 peaks (as ranked by the TRF tool) are similar to each other as shown in the heatmap, where most of them have a high value for the same latent factor (#37).

In **Figure R4**, we also show the DNA sequence for the top five peaks with repeat regions. Further we show peaks of two types in **Figure R5** below, showing the disentanglement done by BindVAE on peaks with low-complexity regions and TF binding sites.

Figure R5 [also **Supplemental Figure S4**]. Latent projection of peaks with only repeat regions and peaks with some low-complexity repeating patterns and TFBS for a TF. This shows the disentanglement performed by BindVAE.

‘Noisy’ dimensions: In addition to the dimensions representing low-complexity regions, there are some dimensions whose k-mer distributions look close to uniform, and projecting DNA sequences onto these gives latent score vectors with uniform values. There are 10 such dimensions in the GM12878 model. See **Figure R6** below for a heatmap showing the projection of GM12878 ATAC-seq peaks onto these noisy dimensions.

Cleavage bias: We constructed motifs using the k-mer distributions from some of the unmapped latent dimensions, but the motifs did not match known Tn5 transposase

binding motifs (although we note there are discrepancies between different published motifs). Early in the project, we applied BindVAE to DNase-seq data and indeed found latent dimensions that seemed to correspond to DNase I cleavage bias (data not shown). Potentially, the k-mer sequence signal associated with Tn5 transposase bias is less pronounced, or this signal has a positional nature that is not well captured by our bag-of-words representation.

Figure R6.[also Suppl Fig S6] GM12878 peaks projected onto the 10 noisy latent dimensions.

- 5 ***In the "BindVAE learns diverse [...]" section, the authors point out that the theta matrix is diagonal-heavy, and use this to suggest that the learned dimensions represent different TFs. It would be informative to see this same matrix generated on genomic background regions, i.e., regions lacking ATAC-seq peaks. The authors should also provide an illustration of dimensions "that capture highly redundant information", and provide some discussion of what those dimensions capture.***

Redundant dimensions: these are groups of dimensions that capture very similar k-mer distributions as shown in **Figure R7** below for HEY1, NFIA and TFAP4, where each TF is assigned to two latent dimensions, both of which have high weights for very similar k-mers. These are potentially due to binding sites for paralogous TFs or TFs from the same subfamily that have highly similar k-mer distributions. The below plots were generated

by picking the top 1000 k-mers from each of the two latent dimensions (e.g. #69 and #83 for HEY1) and showing the weights of the union of these k-mers.

Figure R7. [also **Suppl. Figure S2**] Redundant dimensions. Latent dimensions with similar k-mer distributions are shown. In each plot, the two dimensions were mapped to the same TF; for example: #69 and #83 are both mapped to HEY1. Along the x-axis is the union of the top 1000 k-mers from both dimensions. The values in each row are the decoder parameters learned by the model.

To depict the learned theta matrix (k-mer distribution matrix) that is trained on genomic background regions, we trained two models:

(Model 1) Random regions of naked DNA. Details of dataset used are in **Figure R8a** below.

(Model 2) flanking regions of GM12878 peaks that are 10kb away

As shown in **Figure R8a** below, we find that only a couple of distinct k-mer patterns are learned by Model 1. On the other hand, Model 2 does learn several distinct TF-like signatures similar to the GM12878 model trained on peaks only – please see **Figure R8b** below. We believe this to be the case because distal regions of “promoter peaks” also contain binding sites, and our flanking set of peaks is likely to contain some of these.

Figure R8. Interpreting the models learned on random regions of the DNA. **(Top)** Matrix showing the learned decoder parameter θ (that shows the k-mer distributions for each latent factor) for a model trained on 150,000 random regions of naked DNA from mouse E16.5 sorted germ cells. **(Bottom)** Matrix showing the learned decoder parameter θ for a model trained on flanking genomic regions from GM12878 cells, where regions that are 10kb away from the ATAC-seq peaks were chosen as training data.

- 6 ***In "Motifs discovered de novo[...]", there is reference to "obtain[ing] the top-200 10-mer sequences that scored the highest", but it is not clear how they are calculating the score. Is this simply the value of the latent dimension when the input contains only the 8-mers generated from the candidate 10-mer? Is it some sort of attribution to individual 10-mers? The authors claim that their model "is biased toward learning shorter motifs more accurately.". The authors must demonstrate that their motifs are more accurate than the motifs in CIS-BP.***

We apologize for the confusion and clarify the 10-mer based motif generation process. Further, we mean that the model more accurately learns shorter rather than longer motifs.

- 7 ***Throughout the manuscript there are interpretations surrounding latent dimension "scores" that refer to Algorithm 1. Please clarify in the methods what is considered a "score" and use consistent wording throughout the Methods and the manuscript body so the reader has a clear idea of what is used as latent dimension scoring. Additionally, please explicitly mention whether the TF probes are from the HT-SELEX experiments in the Methods description of Algorithm 1.***

Thanks for pointing this out. We have made this clarification.

- 8 ***In "mapping peaks to TFs", it is not clear why the authors assign each peak to a single TF. As they illustrate, multiple TFs bind to every ATAC peak, and the paper would be strengthened by representing each peak by its ensemble of TF signals instead of its maximal signal.***

We assign a peak to the top 3 TFs. We have corrected the text in the paper.

- 9 ***In the manuscript body under "Projecting HT-SELEX proves into the latent space", the language in paragraph 1 describing the rationale for using HT-SELEX needs to be expanded upon. Earlier in the manuscript, it has been strongly shown that the VAE is able to learn consensus patterns of TF binding motifs (linking VAE latent dimensions <-> discovered motifs). The addition of HT-SELEX is meant to bring in vitro binding TF information into play to validate that these motifs are indeed meaningful to TF binding (linking VAE latent dimensions <-> in vitro binding). This needs to be described more clearly, with the rationale being laid out prior to describing the heatmap of***

Figure 2B. Furthermore, the biological meaning of this finding should be explored. For example, do these similar TFs tend to bind in the same places (implying that the model has correctly mapped them into the same latent dimension since they represent similar biological processes), or do these similar TFs have distinct binding profiles, suggesting that the model has oversimplified its latent space? It would also be informative to compare the learned motifs and PWMs for one of these families of TFs.

To understand whether the groups of TFs from Figure 2b bind in the same places, we look at the ATAC-seq peaks that overlap with ChIP-seq peaks from TCF3 and TCF12 (with overlap of at least 50bp). We find that 76% of TCF3 peaks that overlap with ATAC-seq also overlap with TCF12. Analogously, 56% of TCF12 overlaps with ATAC-seq also overlap with TCF3, indicating a high co-occurrence of binding sites between these two TFs.

Since we do not have ChIP-seq data in GM12878 for all the groups of TFs from Figure 2b in the main paper, we list their CIS-BP motifs below (**Figure R9a**) to illustrate that they are very similar and hence get projected into the same region of the latent space by our model. This figure is part of the **Supplemental Figure S11**.

The learned motif for OLIG3 and the CIS-BP PWM is shown in the main text Figure 1(c). For the TBX sub-family we show the two motifs learned in **Figure R9b**, where we see the patterns from the BindVAE motif come from either the beginning or the end of the 16bp motif of TBX21.

Figure R9a.[also **Suppl Figure S11**] CIS-BP motifs for TFs from the same family or for paralogous TFs are shown, to illustrate the difficulty of learning TF-specific patterns for these. We show the TFs from the heatmap of Figure 2b (TFs in the boxes). Each group of TFs gets projected to the same latent factor by our model as discussed in the main paper.

Two motifs learned for dim #30
which is mapped to TBX group

One motif learned for dim
2, mapped to TBX4

CIS-BP motif for TBX21

Figure R9b. Top row shows three motifs learned by BindVAE for TBX group of TFs and one that is mapped to only TBX4. At the bottom we show the CIS-BP motif for TBX21 for comparison. As can be seen the patterns from the BindVAE motif come from either the beginning or the end of the 16bp motif.

- 10 *In “top 8-mers learned [...]”, the authors should address the role of reverse-complementation in their sequences. They compare GGGGA to TTCCC and claim that these represent different ends of the motif, but they are nearly reverse complements of each other (since the motif is essentially palindromic). The authors should carefully choose a motif that is not palindromic if they wish to claim that a single latent dimension has captured multiple parts of a motif.*

We show some more TFs and the top 8-mers in **Supplemental Figure S9** and updated **Figure 2d** to show the non-palindromic motif of TFAP4 (also in **Figure R10** below). This motif is 10bp long, and we see 8-mers from both the prefix and the suffix part of the motif. We find that for longer binding sites of ~12bp to 15bp, a single latent dimension can capture the constituent k-mer patterns only if the motif is palindromic. For non-palindromic longer binding sites, we find that the binding patterns are split across two latent dimensions.

Figure R10. [also Suppl Fig S9] Top 8-mers for BindVAE latent dimensions representing TFAP4 and CTCF in the GM12878 model. The CIS-BP motif corresponding to the TF that was assigned to each latent dimension (using Algorithm 1) is also shown. Since CTCF is assigned to multiple latent dimensions, the top 25 8-mers from each are shown.

- 11 *In “Cooperative binding signals”, the authors show that one particular dimension seems to capture two TFs. By comparing the activation of that dimension given sequences that bind to the two TFs individually, they show that sequences designed to bind to both simultaneously are more active in that dimension. This claim would be strengthened with appropriate controls on the k-mer distributions of the input sequences. For example, the CAP-SELEX protocol could be more efficient than HT-SELEX, leading to a generally more motif-rich final sequence. The authors should ensure that the degree of enrichment between the two experiments is the same, and they should choose a pair of TFs that have distinct binding motifs. If the two TFs have very similar binding motifs, it is unsurprising that CAP-SELEX would introduce two*

similar motifs into its designed sequence. If the corresponding latent dimension cannot distinguish between the motifs, then it will naturally give a high value when its motif is enriched in the input.

With respect to the hypothesis that the “CAP-SELEX could be more efficient than HT-SELEX, leading to a more motif-rich final sequence”, we caution that the degree of motif enrichment is highly variable between different HT-SELEX and CAP-SELEX experiments, so it is not possible to make a global statement. We do not know of a general procedure to guarantee a comparable level of enrichment. Therefore, we feel it is reasonable to present the CAP-SELEX analysis as an analysis vignette to explore sequence signals captured by the model, rather than as a comprehensive assessment of composite motifs.

All examples of cooperative motifs that we find involve TFs with distinct binding motifs. We now show the CIS-BP motifs for the two TFs in **Figure 3a** and **3b**. To further clarify some of the concerns raised by the reviewer, we show two types of examples in the **Figure R11** (also included as **Supplemental Fig. S10**) below:

- (a) The cooperative binding motif is enriched, while the individual TF motifs are not;
- (b) The individual motif is enriched, while the cooperative motif is not.

Figure R11 [also **Suppl Figure S10**] Examples of CAP-SELEX probes scoring higher and lower than individual TF probes. **(Top)** Example of cooperative binding CAP-SELEX probes of TFAP4:FLI1 being enriched while the individual TF probes are not enriched for TFAP4 or FLI1 ($p = 1$). **(Bottom)** Example where individual TF probes from the SELEX experiment for EOMES are enriched, while CAP-SELEX probes of cooperative binding between MYBL1:EOMES are not.

In these examples, we find that both the individual TF motifs and the cooperative motifs are only enriched if the cooperative binding motif is simply a concatenation of the constituent TF motifs. In **Figure R12**, we list the CIS-BP motifs for the TFs from **Figure R11** above.

Figure R12. CIS-BP motifs for examples from **Figure R11**.

- 12 *In "Accessibility patterns predicted[...]", the authors must demonstrate that their model gives similar results for similar cell types, rather than running two drastically different cell types. It would be reassuring to see two datasets collected for the same cell type by different experimental groups to know that the different activation patterns are due to actual biological differences.*

We now illustrate the difference in activation patterns for similar cell types by training on samples coming from the same cell type, but different donors: one male donor and one female donor for naïve B cells (**Table R2**). We look at the TFs found in both and plot their accessibility scores (**Figure R13**, also in **Supplemental Figure S7**).

The reviewer's question is also directed at the stability of the model's learned representations. Since BindVAE is a probabilistic model that optimizes the likelihood of the data, there is some variance/uncertainty across experimental runs. This is unlike autoencoders since the latent vector here is *sampled* from the Dirichlet posterior distribution. In addition, there is the variance from the randomized initialization of model parameters. We quantify this uncertainty by training models 5 times using identical hyperparameters, on data from only the female donor sample of naïve B cells. We plot the accessibility scores for the union of all TFs found, along with the standard deviation across the 5 repeat runs. While there is variance in the exact TFs found across the repeat runs, if we visualize the k-mer distributions for two of the repeat experiments in **Figure R13**, we see that they are quite similar. Compare this to the k-mer distributions learned for two distinct cell types in **Figure R1** or for the same cell-type but samples coming from different donors.

Experiment	T-cell female adult ^a
replicate 1, number of peaks	79802
replicate 2, number of peaks	82359
overlap ^b in peaks	68682
Experiment	naive B-cell
female donor ^c , number of peaks	87500
male donor ^d , number of peaks	94007
overlap in peaks	76442
Experiment	CD8 T-cells
mouse	221,053
human	190,816

Table R2. Details of additional datasets used.

Figure R13. [also **Suppl Figure S8**]. **(Top left)** k-mer distributions from two models trained on naive B cells from two human donors, male and female. **(Top right)** k-mer distributions from two models from two repeat experiments on female naive B cells. In both cases, there is not much variance in the learned patterns, as these are biologically close samples. **(Bottom left)** k-mer distributions from the two models trained on GM12878 and A549. Since these are distinct cell types, we see two distinct manifolds in the patterns being learned. There is also a cluster of latent dimensions at the bottom of the plot that captures similar k-mer patterns. **(Bottom right)** k-mer distributions from four models: GM12878, A459, mouse CD8 T-cells and human CD8 T-cells.

Figure R14. [also **Suppl. Figure S7**] Presence of binding sites as indicated by the ‘accessibility score’ is shown in two settings. **(Left)** Two different samples of naïve B cells are shown: a male donor and a female donor. **(Right)** On the female sample, 5 models were trained using the same hyper-parameter settings to get 5 repeat runs over the same input data. The learned TFs are shown along with the averaged accessibility score across 5 experiments and the standard deviation.

- 13 *In "Comparison with HOMER", the authors should assess the quality of the motifs identified by HOMER. As seen in Figure 1, the motifs from BindVAE are often different than the HT-SELEX PWMs. Are the HOMER motifs more similar to the HT-SELEX ones? It would also be good to compare motifs derived from other, more recent, approaches.*

Figure 1 shows CIS-BP motifs from a variety of sources (HT-SELEX or ChIP-seq or other PBM arrays). We add another column to the table in **Figure 1c** that shows the HOMER motif (if found by HOMER), for each row's TF. This table is shown on the next page here for reference (**Table R3**).

We also compare to two other approaches: MEME, GADEM. The goal of the comparisons with the motif-discovery approaches is to illustrate the ability of BindVAE to find more number of distinct motifs as is indicated by the higher recall numbers in **Supplemental Table S3** (see also **Table R6** below).

Latent dimension	Assigned TF	Learned motif	CIS-BP motif for TF	HOMER motif (if found)
72	NFKB1			
7	TFAP4			
81	SPI1			—
60	RUNX3			
39				
82	OLIG3			—
91	SMAD3/ PKNOX1			
30	EOMES			—
79	HNF1B			—
11	CTCF			
4				
25	PRDM1			
51	RXRB			—
54	VDR			—
6	GATA4			
52	ETS1			—

Table R3. [also Figure 1c] Motif analysis of each latent dimension shows that different binding specificities are captured by each latent factor in GM12878. Column 2 shows the name of the TF assigned by Algorithm 1, column 3 shows the CIS-BP motif corresponding to the TF and the last column shows the motif that was found by HOMER using de novo motif discovery for the TF.

- 14 **The methodology of how the "accessibility score" is computed in the manuscript body (linked to Figure 3D) should be moved to the Methods section. Additionally, this segment on accessibility patterns would be strengthened by deeper discussion into the findings. This is an opportunity to explore differences between cell types in order to biologically validate your computational methods of de novo motif discovery. Do these results make sense? For TFs with similar accessibility scores, do they contain similar consensus k-mer reconstructions or are there differences?**

We pick some TFs with the same accessibility scores in GM12878 and A549, as seen in **Fig. 3d** in the main text (please also see **Table R1** above). For these, we show the motifs obtained by our approach below (**Table R4**). We see differences in the relative importance of nucleotides at multiple positions if we compare the motifs found for GM12878 and A549. The CIS-BP motifs for each of these TFs is shown in the last column for reference.

TF	GM12878 motif	A549 motif	CIS-BP motif
HNF4A			NFIA			SRY			
Table R4. [also **Suppl. Table S1**] Motifs constructed from latent factors learned for GM12878 and A549 for TFs learned by our model. HNF4A, NFIA, SRY have the same 'accessibility score' in both GM12878 and A549 trained models (see Figure 3d in the main paper).

- 15 **In the discussion, the authors claim that CNN-based methods do not generalize across cell types. It is not clear that the method in this paper is able to generalize across cell types short of being re-trained on each different cell type. For example, if there is a TF that is found in cell type B, but not cell type A, and the model is trained on type A, there is no expectation that the B-specific TF will be found. CNN-based models must, by their nature, be trained on every cell type of interest, and this is a limitation of**

these methods. If the authors have developed a method that can be trained once and then allow for interpretation of data from different cell types, they should clearly illustrate this useful finding.

We apologize for the confusion. We were trying to make the point that while it is possible to train discriminative models (including CNNs, string kernel SVMs, and others) on TF ChIP-seq data in the few cell types where such TF occupancy data is plentiful, these models do not generalize well to other cell types. Therefore, these discriminative methods have proved to be of limited use for predicting TF occupancy across ATAC-seq peaks in a new cell type based on training models e.g. on ENCODE cell line ChIP-seq data. We propose instead to apply BindVAE directly to ATAC-seq data in the new cell type. We have tried to clarify these points in the Introduction.

- 16 ***Figure 4A and B would be strengthened by also showing the different TF binding motif consensus for each TF classification. This suggestion applies particularly to SPI1 and IRF4 in Figure 4B due to the poorer clusters formed by UMAP. Are these two motifs highly similar? Why is IRF4 having trouble being distinguished? Are these motifs highly cooperative? What is the overlap of ChIP-seq peaks themselves (i.e. are IRF4 and SPI1 ChIP-seq peaks overlapping in vivo as well? Are the latent dimension ambiguities shown by poor UMAP consensus clusters reflecting something we see in vivo as well?)***

Through analysis of ChIP-seq data for IRF4 and SPI1 in GM12878, we find that there is little overlap (34 out of 1000 peaks overlap) between peaks of these two factors. This is possibly because IRF4 binds DNA weakly but cooperative binding with factors such as SPIB in B cells increases binding affinity [Li et al., 2012].

- 17 ***Figure 4C is missing axis labels. Also currently the authors are claiming "The latent dimensional scores based on ATAC-seq enriched k-mers can distinguish different TF binding peaks." However, this is not strongly conveyed, because the reader does not know which TF ChIP-seq peak is connected to which row. Presumably, we can assume that the top 66% of rows are attributed to RUNX3, but we can't be sure because that information is not available. If you can add an extra column/annotation that can connect the row with its TF identity, this figure would be even stronger. This is particularly true because it appears that NRF1 seems to contain co-binding relationships with RUNX3 and MAX in some cases, but since we cannot be sure of each TF's identity, then this is difficult for the reader to assume.***

We have added the axis labels and thank the reviewer for catching this issue. To clarify which TF ChIP-seq experiment is connected to which row, we have added a second

heatmap with the CHIP-seq ground truth label assigned to the ATAC-seq peak via overlap (i.e. a minimum of 50bp overlap between a CHIP-seq and ATAC-seq peak to assign the TF label) in the same order as **Figure 4c**. Please refer to **Figure R15** below (also in **Supplemental Figure 14**). This will allow the reader to get the **annotation** information for each row. We observe the co-binding relationship between NRF1 and MAX as pointed out by the reviewer. We do not however, see co-binding between NRF1 and RUNX3.

Figure R15. [also **Suppl. Figure S14**] Matrix of GM12878 ATAC-seq peaks ordered as per the matrix ranked by latent factor scores (topic scores) from Figure 4c. Rows are peaks and columns are TFs. Each entry (i, j) in the i -th row and j -th column of the matrix is the i -th peak's CHIP-seq annotation for TF j .

Reviewer #3:

- 1 ***A detailed and mathematically clear description of the model is important. To avoid ambiguous text sections in the Methods section, we suggest introducing clear notations, formulas for the definitions like model loss, input vector of the model, and output vector.***

We have revised the Methods section to add the requested definitions and information to the paper.

- 2 ***The authors claim that some latent dimensions recover cooperative binding. "Since the co-occurrence of multiple binding patterns might merely suggest that binding sites of two different TFs are present in a peak and not necessarily cooperative binding" the authors use sequences from CAP-SELEX experiments and show that the activation the latent dimension of interest is higher for CAP-SELEX sequences than for the HT-SELEX sequences from the two separate TFs. We have the following concerns. First, the procedure of how the two shown examples are chosen is not clear. What made dimension 60 and 67 distinct? This feels cherry-picked. Any overall statistics?***

The two examples were randomly picked out of the 13 total composite motifs that we find over the 100 dimensions in the latent layer. All 13 pairs of TFs are listed in **Supplemental Table S5** (please refer to **Table R5** following Q5 below). The probes from these 13 pairs of TFs had significantly higher scores than those of other pairs from CAP-SELEX experiments or single-TF HT-SELEX experiments.

Combinations of TFs with successful CAP-SELEX experiment are limited after filtering for quality metrics as described in our BindSpace paper (Nature Methods 2019). Even after filtering for QC, in only a handful (~70 of them) is there a composite motif that differs from the concatenation of individual motifs for which we can assess enrichment in the latent dimensions. Rather than make systematic statements about composite motifs given the sparsity of CAP-SELEX data, we are only presenting "analysis vignettes" that make the point that in some cases the latent dimension can capture a composite motif.

- 3 ***Second, the authors need to prove that the higher score is not merely because the CAP-SELEX sequences has both motifs in it. So the adequate comparison would be CAP-SELEX sequences compared to sequences which contain the motifs of both TFs.***

See **Supplementary Figure S10** (**Figure R11** above) where the CAP-SELEX probes are not enriched for EOMES_MYBL1 while the individual TF's SELEX probes are enriched for EOMES (the p-value for the Wilcoxon signed rank test for enrichment is shown along the

x-axis). This shows that CAP-SELEX probes do not intrinsically achieve higher latent scores due to both motifs being present.

Also, some composite motifs contain one or both of the independent TF motifs, while others do not match either motif and the composite pattern is a modified variant of one or both of the original individual TF motifs. This is illustrated in **Figure 3a** in the main paper and **Supplementary Figure S10 (Figure R11 above)**, where individual TF motifs are not enriched for TFAP4 and FLI1 while the composite motif for TFAP4_FLI1 is enriched.

4 **Third, the authors compare the activation of FOXJ2 and FOXL1 to FOXJ3-TBX21 and not FOXJ3.**

We do not show FOXJ3 as it is not enriched in dimension #67, i.e. the enrichment has a p-value $p = 1$.

5 **Fourth, the authors show an example of how the cooperative binding motif looks for FOXJ3-TBX21 but not for MYBL1-MAX.**

We have added the motif for MYBL1-MAX to **Supplementary Table S5 (Table R5 below)**. We also show some other TF pairs, i.e. cooperative binding motifs found by BindVAE.

	Other cooperative TF pairs found	
ELK1_SPDEF	ELK1_TCF15	MYBL1_MAX
ELK1_NFKB1	TFAP4_FLI1	TFAP4_MAX
E2F1_NHLH1	ETV2_NHLH1	HOXB2_NHLH1
POU2F1_FOXO6	E2F1_ELK1	MYBL1_MAX
FOXJ3_TBX21		

Table R5. [also **Suppl. Table S5**] MYBL1:MAX motifs computed from the GM12878 model are shown in the top row. The bottom table shows all co-operative binding pairs of TFs found for GM12878 by our model using the CAP-SELEX data are shown.

- 6 ***The authors claim that they can predict accessibility patterns for GM12878 and A549 by using the summed activation over all peaks for each latent dimension. While it is not surprising that the latent activations differ between cell types (as they have been trained on separate sets of ATAC-peaks) there is also no comparison to what factors are known to be active in those cell types.***

To compare with active TFs, we show the correlation of gene expression and accessibility patterns for GM12878 below. Expression of TFs at the RNA level is an imperfect but useful proxy for their activity in a given cell type. The RNA-seq data used for these experiments is described in the “Datasets” section of the paper.

For each cell type, the scatter plots show the correlation between the TF accessibility scores from our model (or motif hits % from HOMER) and expression values (see **Figure R16** below). Note that this plot does not include TFs that are found by the model but not mapped to a TF name due to the limited set of 270 HT-SELEX TFs that we use (there are 29 such TFs for GM12878). It also does not include non-SELEX TFs that are expressed but not found by our approach.

We find that the Pearson correlation coefficient is low for A549 and is very low for HOMER in general. An issue with correlation computations is -- each method (BindVAE / HOMER) returns multiple motifs for a single TF with a different 'accessibility score' / % target hits. We take the maximum over all the candidates. Even if we consider TFs at the subfamily level, some aggregation needs to be done. So, a groups of motifs with their accessibility-scores needs to be matched to a group of genes with their RNA-seq expression values and using the *maximum* value over a group is not an accurate assumption about the expression data and possibly leads to the poor correlation seen.

BindVAE: GM12878 correlation between accessibility score and gene expression (RNA-seq)

HOMER: GM12878 correlation between motif hits % and gene expression (RNA-seq)

BindVAE: A549 correlation between accessibility score and gene expression (RNA-seq)

HOMER: A549 correlation between motif hits % and gene expression (RNA-seq)

Figure R16. Scatter plots showing the correlation between RNA-seq based TF expression and the accessibility score obtained from BindVAE for GM12878 (**top**) and A549 (**bottom**). The (**middle**) figure shows this correlation for HOMER, where we use the % motif hits as a surrogate to 'accessibility score'.

- 7 ***The authors compare their identified TFs to algorithm HOMER. One issue is that HOMER fits PWMs and then does a matching against a provided set of PWMs. It is unclear whether the difference between HOMER and BindVAE comes from the PWM fitting algorithm, which corresponds to the fitting of the autoencoder, or from the PWM matching algorithm, which corresponds to BindVAE's post-processing Algorithm 1. The authors should provide an analysis that delineates both contributions.***

In order to delineate the contribution of Algorithm-1 vs fitting the VAE, we replace Algorithm-1 with Tomtom and a new comparison with BindVAE+Tomtom. We also compare with GADEM and MEME, each of which only discovers novel motifs. For both GADEM and MEME we use Tomtom for the PWM-matching component. So across these three methods, we first run the motif discovery component of each approach and then use Tomtom for TF-matching.

We find that the results for BindVAE with Tomtom are comparable to those obtained with Algorithm-1. All results are listed in **Supplementary Table 3a, 3b** (please see the responses to question 8 and 9 on the next few pages).

- 8 ***A second issue with the comparison with HOMER is that it is done for fixed cutoffs. The authors should provide instead "continuous" comparisons, e.g. using ROC or PRC curves.***

All the motif finding algorithms and our TF-matching algorithm (Algorithm 1) assign p-values to the matches, which are not appropriate for use as 'scores' from a classifier for generating PRC curves. That is, each point on a PRC curve for HOMER would come from a different 'model', whereas for BindVAE all points would come from the same model as the TF-matching Algorithm 1 is a post-processing step. Hence these curves would not be comparable. We instead show results for BindVAE and HOMER, for a few different p-value / e-value cut-offs and show the results in **Table R6a** and **Suppl Table S3(b)**.

HOMER: The way to achieve more comparisons with HOMER is by changing the e-value cut-off (the default value is: 0.01). This requires re-running the entire algorithm, which is also the case for other motif discovery approaches.

BindVAE: being a machine-learning model only requires to be trained once. The p-value changes need to be made in the post-processing in Algorithm 1, which takes ~5 mins to complete. There are two thresholds that can be changed:

- the p-value cut-off on line-8 of Algorithm 1 or
- the top % of all probes where we look for enrichment of a TF's probes on line-7. We try the following three values: 3%, 5%, 10%

BindVAE + Tomtom: We also compare with a version of BindVAE, where we replace Algorithm-1 with a PWM matching approach like Tomtom that takes PWMs generated for each latent dimension and matches them against a database. Here also, two thresholds can be changed: the e-value cut-off of Tomtom or the parameters of the MEME algorithm, which we use to generate PWMs from 10mers. We do not change the MEME parameters however and only modify the e-value threshold of Tomtom.

All positives classifier: The performance of a naive match-every-motif classifier (label everything positive) is shown. This gives a sense of the class-skew.

- 9 ***A third issue is that the comparison is limited to only one motif finding algorithm. A review of existing motif enrichment algorithms should be done and the most prominent ones should be compared to. I am expecting MEME, GLAM2, and probably GADEM. Eventually, the performance should be put in perspective of overall compute time for the different methods.***

We compare with the following approaches and report the results in **Table R6b** (also **Suppl Table S3a, S3b**).

MEME: We ran MEME in tandem with Tomtom, where the learned motifs from MEME are used as input to Tomtom which matches the learned motifs with the PWMs from our dataset of 270 HT-SELEX TFs. We used the default parameters to run MEME, except for the following: number of motifs=500, background-model=2nd order, motif distribution=anr, no e-value threshold. The following command was used:

```
./meme -dna -mod anr -nmotifs 500 -minw 5 -maxw 15 -markov_order 2 -p 10  
<fasta-file>
```

MEME was run with 10 processors in parallel. On both datasets, MEME+Tomtom took 10 to 12 hours to run, with Tomtom taking under 5 minutes.

GLAM2: GLAM2 is an approach for creating an alignment given a small number of sequences and generating the subsequent motif from the MSA. Hence, it did not seem like an appropriate algorithm to compare against.

GADEM: Overall, this genetic algorithm guided approach is non-deterministic (produces differing number of motifs in each run) and suited for learning a single TF's motif/PWM from ChIP-seq data. Since GADEM can only find motifs, we ran it in conjunction with Tomtom for motif-matching to find which SELEX PWMs are found.

The R-package rGADEM had some limitations (can only work with 44000 sequences, limited number of motif matches etc.); we modified the source code as per the

documentation. We used the default parameters (e-value threshold=0), except for the number of motifs to find, which we set as: nmotifs=500. We tried several p-value cut-offs, including the recommended one (0.0002) and less stringent cut-offs. Relaxing the p-value cut-off to values higher than 0.001 results in millions of motif matches being found and the code segmentation faults. Oddly, fewer motifs were found for the less stringent cut-offs of 0.001, 0.0008.

The algorithm had run times between 2 to 7 hours (with lower times for more stringent p-value cut-offs). With a cut-off of 0.0002, 4 to 8 motifs were found ranging in length from 9 to 31.

Method	Run-time	GM12878			A549			T-cells female adult		
		Prec.	Rec.	F1	Prec.	Rec.	F1	Prec.	Rec.	F1
GADEM+Tomtom	2-7 hrs	0.634	0.154	0.247	0.571	0.090	0.154	0.760	0.186	0.298
MEME+Tomtom	9-10 hrs	0.530	0.261	0.348	0.791	0.327	0.462	0.686	0.361	0.472
HOMER	6-24 hrs	0.608	0.351	0.444	0.765	0.504	0.607	0.659	0.811	0.728
BindVAE	4-5 hrs	0.636	0.416	0.502	0.806	0.359	0.496	0.785	0.351	0.485
BindVAE+Tomtom	4-5 hrs	0.573	0.952	0.714	0.733	0.900	0.807	0.638	0.821	0.718

(a)

Method	p-value / e-value	GM12878			A549			T-cells female adult		
		P	R	F1	P	R	F1	P	R	F1
All positive classifier		0.562	1.000	0.720	0.736	1.000	0.828	0.628	1.000	0.771
HOMER	0.001	0.631	0.386	0.478	0.783	0.377	0.508	0.780	0.414	0.540
HOMER	0.01	0.608	0.351	0.444	0.765	0.504	0.607	0.659	0.811	0.728
HOMER	0.1	0.628	0.452	0.525	0.777	0.350	0.481	0.748	0.505	0.602
HOMER	10	0.719	0.458	0.559	0.800	0.418	0.548	0.764	0.430	0.550
BindVAE	3%, 0.005	0.744	0.208	0.324	0.833	0.113	0.198	0.821	0.164	0.273
BindVAE	5%, 0.05	0.636	0.416	0.502	0.806	0.359	0.496	0.785	0.351	0.485
BindVAE	5%, 0.1	0.636	0.541	0.584	0.779	0.513	0.618	0.766	0.436	0.554
BindVAE	10%, 0.5	0.579	0.803	0.672	0.747	0.754	0.750	0.721	0.611	0.661
BindVAE	10%, 0.8	0.561	1.000	0.718	0.735	1.000	0.847	0.765	0.382	0.509
BindVAE+Tomtom	0.1	0.533	0.047	0.086	0.750	0.040	0.075	0.541	0.038	0.07
BindVAE+Tomtom	1	0.539	0.410	0.464	0.737	0.204	0.318	0.602	0.481	0.534
BindVAE+Tomtom	10	0.573	0.952	0.714	0.733	0.900	0.807	0.633	0.796	0.705
BindVAE+Tomtom	20	0.571	0.976	0.720	0.731	0.954	0.827	0.638	0.821	0.718

(b)

Table R6 Precision, Recall, F1 achieved by the various de novo motif discovery approaches in retrieving TFs from ATAC-seq peaks of the two cell types. Expressed TFs (from RNA-seq data) that intersect with our HT-SELEX set of TFs are used as the gold-standard for retrieval (a) Performance with default thresholds are shown (b) For the two best approaches, HOMER and BindVAE, performance upon varying the cut-offs is shown.

BindVAE + Tomtom: We also compare with a version of BindVAE, where we replace Algorithm-1 with a PWM matching approach like Tomtom that takes PWMs generated for each latent dimension and matches them against a database.

We find that BindVAE has a higher precision compared to other approaches, but if we replace the TF-matching procedure (from Algorithm-1) with Tomtom, we find that this results in a higher recall and F1. HOMER has a good performance on the T-cell dataset.

10 ***A fourth issue is that the Methods section provides no detail on how HOMER was applied.***

We have added this information to the Methods section. We first process the set of HT-SELEX PWMs from Jolma et al. into a database format that is used as input to HOMER.

HOMER was run in the *de novo* motif discovery mode by invoking the perl script findMotifsGenome.pl in the following manner:

```
./findMotifsGenome.pl <peak-file.bed> [hg19/hg38] <output-dir> -S 1000 -p 10 -size  
given -len 6,8,10,12 -noknown -mset jolma
```

11 ***With Figure 4's analysis, the authors aim to show that the latent dimensions correspond to known TFs by using CHIP-seq peaks for the identified TFs. The authors find that only 12 TFs overlap between HT-SELEX and reliable CHIP-seq data but do not specify what a reliable CHIP-seq data means in that context (all of them come from ENCODE).***

We only downloaded CHIP-seq data which has a green "Audit category". We excluded experiments with "Audit category" = orange or red as these have insufficient read length (e.g. POLR2A), insufficient read depth (e.g. ZNF274), poor library complexity (e.g. NFATC3), partially characterized antibody (e.g. SMAD1). This gave us 82 experiments.

12 ***Furthermore, the representation chosen (Figure 4c) to prove the claim is not nailing the point. One could think of plotting a barplot where they compare latent activations of peaks with CHIP-seq signal to all other peaks. Alternatively, one could also phrase it as a prediction task by evaluating how well a latent dimension can distinguish peaks with that TF bound (overlap with CHIP-seq) from all other peaks. Evaluation metric would be AUPRC.***

To address this, we show a heatmap that has the same row ordering as the heatmap in **Figure 4c**. For each ATAC-seq peak (i.e. for each row), we show the TF CHIP-seq peaks that it overlaps with. Please see **Supplemental Figure S14 (Figure R15 above)** for full details.

Second round of review

Reviewer 1

The concerns are addressed well.

Reviewer 2

Kshirsagar et al. have significantly expanded on the manuscript and have addressed most of my concerns. I appreciated the authors' very detailed and clear responses to the initial comments and their highlighting the relevant changes in the manuscript. This made this second round of reviewing not only easy, but downright pleasant. The paper is now much stronger with more validation and checking of the motifs. It still did not comprehensively take advantage of the combinatorial/cooperative nature of how the VAE interprets bag-of-words k-mers, but the Supplementary Figure 10 does provide evidence that the CAP-SELEX does not ALWAYS show higher enrichment. The discussion and expansion of methodology and possible limitations now makes sense, and I am satisfied that they are considering repeats, palindromes, multiple latent spaces for longer motifs, and showing negative controls of training adequately. I am particularly happy with the expansion of SFigs to show the latent space representations when trained on random/distal regions as well as using B/T cells from male/female donors to show differences/similarities across different iterations of model training and highlight similar cell type results from different cell type results.

Taken together, this is a nice example of unsupervised learning applied to ATAC-seq data and should be published.

Minor comments:

In point 14, I requested biological analysis of the different TF distribution in different cell types. The authors have not commented on this (though they have addressed the other comments in point 14), and it would be nice to see something as simple as a GO term analysis showing that the TFs present in these different cell lines correspond to the biological processes present in those lines. This is not critical, but it would strengthen the paper, even as just a note in the caption of a supplemental figure.

Figure 2C and 2D both claim that the motif was assembled through the 8-mers of two latent spaces. These figures should have better labeling clarifying which of the 8-mers belong to which latent space.

With all the text additions, the paper should be checked for flow before publication. For example, in the section "BindVAE: a Dirichlet [...]" in the Results section, one paragraph starts with "We use k-mers with wildcards [...]", and the next paragraph picks up with "We use 8-mer (with wildcards) count vectors as input."

Reviewer 3

The manuscript has improved but there are still several points to be addressed.

#1. (mathematically clear description)

R3: Thanks for adding the variable definitions. However, the reconstruction loss remains unclear. Specifically the likelihood function $p_{\theta}(x_i | z_i)$ remains to be specified. Is it based on a count distribution, a Gaussian approximation, etc. ?

#2. (cherry-picked examples of cooperative binding)

R3: The authors provides an insightful response regarding the CAP-SELEX dataset that readers would benefit from. I strongly suggest to integrate this argumentation in the manuscript as other readers may get the same impression of looking at cherry-picked examples.

#4. Third, the authors compare the activation of FOXJ2 and FOXL1 to FOXJ3-TBX21 and not FOXJ3.

R3: This is still confusing. In Fig3a MYBL1 and MAX are shown even though the enrichment p-value is $p=1$, what is different here?

#6. (Comparison between inferred activities and independent proxies of TF activity)

R3: The authors are not addressing the point appropriately and did not integrate their analysis in the manuscript. The association with RNA expression of TFs have limitations indeed. However, this is the only independent data supporting the inferred activities. I therefore recommend that this analysis is integrated into the paper and not only as a response to the reviewers. This analysis should be improved. One limitation with the proposed analysis is that FPKMs are not comparable between TFs. The authors may consider instead scatterplotting fold changes of FPKM vs differential accessibility scores (investigating log-scales on either axis as suited). Similarly, the percentage of HOMER motif hits among peaks is not a good metrics to correlate with activity. Some TFs have more targets across the genome than others. This does not make them more active than other TFs. I suggest to instead scatterplot (log-)fold changes of FPKM vs. differential frequencies (probably itself on a logarithmic or on a log-odd scale, as suited).

#9. (Comparisons to more motif enrichment algorithms)

R3: The authors have addressed the point by performing an extensive benchmark. However, the conclusions are not appropriately reflected in the text.

BindVAE+TomTom has noticeably higher recall and F1 scores than just BindVAE according to your table. It appears that algorithm 1 trades off higher precision for lower recall. You write “We find that BindVAE has a higher precision as compared to the other motif-discovery approaches”, also in the abstract. To accurately summarise the table, one should add “but often worse recall and F1 score”. We appreciate that there are use-cases where precision may be more important than recall or F1, so the higher precision (at lower compute time) can be valuable, but this should be explicitly argued (and the failure cases not hidden in a supplementary table).

#12. Furthermore, the representation chosen (Figure 4c) to prove the claim is not nailing the point. One could think of plotting a barplot where they compare latent activations of peaks with

ChIP-seq signal to all other peaks. Alternatively, one could also phrase it as a prediction task by evaluating how well a latent dimension can distinguish peaks with that TF bound (overlap with ChIP-seq) from all other peaks. Evaluation metric would be AUPRC.

R3: The authors have improved the labeling of heatmap Fig 4c, although there are still no color scale legend (Same for the new Supplementary figure S14). More importantly, having two heatmaps still does not allow easily relating the inferred activities to ChIP seq data. Readers are bound to play the “7 error games” between Fig. 4c and the new supplementary figure S14. The authors should consider one of the earlier suggestions: for each TF a ROC curve or boxplots of inferred activities for ChIP-seq bound vs not bound for each TF (I had suggested a barplot but this was not ideal).

Authors’ response

We would like to thank the editor and reviewers for their excellent suggestions and comments, that have greatly helped this work mature. We are also thankful for their vote of confidence in accepting this work for publication.

Please see below for responses to the latest round of comments.

Reviewer-1:

We would like to thank the reviewer for perusing the revised document and our response to their comments.

Reviewer-2:

We would like to thank the reviewer for perusing the revised document and our response to their comments.

1. **GO term analysis:** *I requested biological analysis of the different TF distribution in different cell types. The authors have not commented on this (though they have addressed the other comments in point 14), and it would be nice to see something as simple as a GO term analysis showing that the TFs present in these different cell lines correspond to the biological processes present in those lines. This is not critical, but it would strengthen the paper, even as just a note in the caption of a supplemental figure.*

Since all proteins involved in this study are transcription factors, we generally see an enrichment of GO terms relevant to various types of protein-binding, transcription factor activity for all cell types. If we analyze the distinct TFs found for each cell type, we do see cell-type specific differences and we have incorporated this text in the manuscript.

2. *Figure 2C and 2D both claim that the motif was assembled through the 8-mers of two latent spaces. These figures should have better labeling clarifying which of the 8-mers belong to which latent space.*

We have clarified this in the Figure caption.

3. *With all the text additions, the paper should be checked for flow before publication. For example, in the section "BindVAE: a Dirichlet [...]" in the Results section, one paragraph starts with "We use k-mers with wildcards [...]", and the next paragraph picks up with "We use 8-mer (with wildcards) count vectors as input."*

We have rectified this.

Reviewer-3:

We would like to thank the reviewer for perusing the revised document and our response to their comments.

1. *R3: Thanks for adding the variable definitions. However, the reconstruction loss remains unclear. Specifically, the likelihood function $p_{\theta}(x_i | z_i)$ remains to be specified. Is it based on a count distribution, a Gaussian approximation, etc. ?*

We have included further definitions

2. *R3: The authors provides an insightful response regarding the CAP-SELEX dataset that readers would benefit from. I strongly suggest to integrate this argumentation in the manuscript as other readers may get the same impression of looking at cherry-picked examples.*

We integrate this text in the manuscript

3. *Third, the authors compare the activation of FOXJ2 and FOXL1 to FOXJ3-TBX21 and not FOXJ3.*

We have added FOXJ3 to the box plot now and it has a p-value of 1. FOXJ3 does not get assigned to any latent dimensions (as it has high p-values for all latent dimensions).

4. *This is still confusing. In Fig3a MYBL1 and MAX are shown even though the enrichment p-value is $p=1$, what is different here?*

Our goal is to show that while individual TF's probes from HT-SELEX may not be enriched, the CAP-SELEX probes for the TF-pair are enriched. We include the individual un-enriched TF's p-values to illustrate this. We clarify this in the manuscript.

5. *R3: The authors have addressed the point by performing an extensive benchmark. However, the conclusions are not appropriately reflected in the text. BindVAE+TomTom has noticeably higher recall and F1 scores than just BindVAE according to your table. It appears that algorithm 1 trades off higher precision for lower recall. You write "We find that BindVAE has a higher precision as compared to the other motif-discovery approaches", also in the abstract. To accurately summarise the table, one should add "but often worse recall and F1 score". We appreciate that there are use-cases where precision may be more important than recall or F1, so the higher precision (at lower compute time) can be valuable, but this should be explicitly argued (and the failure cases not hidden in a supplementary table).*

We add text to the abstract and main text to reflect this

6. *#12. Furthermore, the representation chosen (Figure 4c) to prove the claim is not nailing the point. One could think of plotting a barplot where they compare latent activations of peaks with ChIP-seq signal to all other peaks. Alternatively, one could also phrase it as a prediction task by evaluating how well a latent dimension can distinguish peaks with that TF bound (overlap with ChIP-seq) from all other peaks. Evaluation metric would be AUPRC.*

As suggested by the reviewer, supplementary figures **S17** and **S18** show box-plots depicting the distribution of latent scores for ATAC-seq peaks that are TF-bound (defined by ChIP-seq overlap with the ATAC-seq peak), and not TF-bound. Each plot corresponds to one TF, with plots for all TFs from Figure 4c. We already show results with Precision, Recall, F-score in the main paper for the chip-seq prediction problem suggested by the reviewer.

7. *R3: The authors have improved the labeling of heatmap Fig 4c, although there are still no color scale legend (Same for the new Supplementary figure S14). More importantly, having two heatmaps still does not allow easily relating the inferred activities to ChIP seq data. Readers are bound to play the "7 error games" between Fig. 4c and the new supplementary figure S14. The authors should consider one of the earlier suggestions: for each TF a ROC curve or boxplots of inferred activities for ChIP-seq bound vs not bound for each TF (I had suggested a barplot but this was not ideal).*

We now include supplementary figure S14 as a sub-plot in Figure 4c.